**Quantifying the relative importance of greenhouse gas emissions from current and future**
**savanna land use change across northern Australia**
Bristow M.[1,2], Hutley L.B.[1], Beringer J.[3], Livesley S.J.[4], Edwards A.C.[1], Arndt S.K.[4]
[1]School of Environment, Research Institute for the Environment and Livelihoods, Charles Darwin
University, NT, Australia, 0909
[2]Department of Primary Industry and Fisheries, Berrimah, NT, Australia, 0828
[3]School of Earth and Environment, The University of Western Australia, Crawley, WA, Australia,

11  6009

[4]School of Ecosystem and Forest Sciences, The University of Melbourne, Burnley, Victoria,
Australia, 3121
*Correspondence to*: Lindsay B. Hutley (lindsay.hutley@cdu.edu.au)
**Keywords**: Deforestation, clearing, eddy covariance, savanna burning, fire emissions
**Abstract**
Clearing and burning of tropical savanna leads to globally significant emissions of greenhouse
gases (GHG), however there is large uncertainty relating to the magnitude of this flux. Australia's
tropical savannas occupy the northern quarter of the continent, a region of increasing interest for
further exploitation of land and water resources. Land use decisions across this vast biome have the
potential to influence the national greenhouse gas budget. To better quantify emissions from
savanna deforestation and investigate the impact of deforestation on national GHG emissions, we
undertook a paired site measurement campaign where emissions were quantified from two tropical
savanna woodland sites; one that was deforested and prepared for agricultural land use, and a
second analogue site that remained uncleared for the duration of a 22 month campaign. At both
sites, net ecosystem exchange of $CO_2$ was measured using the eddy covariance method.
Observations at the deforested site were continuous before, during and after the clearing event,
providing high resolution data that tracked $CO_2$ emissions through nine phases of land use
change. At the deforested site, post-clearing debris was allowed to cure for six months and was
subsequently burnt, followed by extensive soil preparation for cropping.
During the debris burning, fluxes of $CO_2$ as measured by the eddy covariance tower were
excluded. For this phase, emissions were estimated by quantifying on-site biomass prior to
deforestation and applying savanna-specific emission factors to estimate a fire-derived GHG
emission that included both $CO_2$ and non-$CO_2$ gases. The total fuel mass that was consumed during
the debris burning was 40.9 Mg C ha$^{-1}$ and included above- and below- ground woody biomass,
course woody debris, twigs, leaf litter and $C_4$ grass fuels. Emissions from the burning were added to
the net $CO_2$ fluxes as measured by the eddy covariance tower for other post-deforestation phases to
provide a total GHG emission from this land use change.
The total emission from this savanna woodland was 148.3 Mg $CO_2$-e ha$^{-1}$ with the debris
burning responsible for 121.9 Mg $CO_2$-e ha$^{-1}$ or 82% of the total emission. The remaining emission
was attributed to $CO_2$ efflux from soil disturbance during site preparation for agriculture (10% of
the total emission) and decay of debris during the curing period prior to burning (8%). Over the
same period, fluxes at the uncleared savanna woodland site were measured using a second flux
tower and over the 22 month observation period, cumulative NEE was a net carbon sink of -2.1 Mg
C ha$^{-1}$, or -7.7 Mg $CO_2$-e ha$^{-1}$.

6       Estimated emissions for this savanna type were then extrapolated to a regional scale to 1)

provide estimates of the magnitude of GHG emissions from any future deforestation and 2)
compare with GHG emissions from prescribed savanna burning that occurs across north Australian
savanna every year. Emissions from current rate of annual savanna deforestation across north
Australia was double that of reportable (non-$CO_2$ only) savanna burning. However, if the total GHG
emission is accounted, $CO_2$ plus non-$CO_2$ emissions, burning emissions are an order of magnitude
larger than that arising from savanna deforestation.  We examined a scenario of expanded land use
that required additional deforestation of savanna woodlands over and above current rates. This
analysis suggested that significant expansion of deforestation area across the northern savanna
woodlands could add an additional 3% to Australia's national GHG account for the duration of the
land use change. This bottom-up study provides data that can reduce uncertainty associated with
land use change for this extensive tropical ecosystem and provide an assessment of the relative
magnitude of GHG emissions from savanna burning and deforestation. Such knowledge can
contribute to informing land use decision making processes associated with land and water resource
development.

# 1.0 Introduction

An increase in greenhouse gas (GHG) emissions through human-related activities is leading to rapid change in the climate system (IPCC 2013). It is therefore crucial to obtain data describing the net GHG balance at regional to global scales to better characterise anthropogenic forcing of the atmosphere (Tubiello et al., 2015). Emissions from land-use change (LUC) are the integral of ecosystem transformations that can include emissions from deforestation and conversion to agriculture, logging and harvest activity, shifting cultivation, as well as regrowth sinks following harvest and/or abandonment of previously cleared agriculture lands (Houghton al. 2012). At present, LUC emits $0.9 \pm 0.5$ Pg C $y^{-1}$ to the atmosphere, which is approximately 10% of anthropogenic carbon emissions (Le Quéré et al., 2014). Data sources and methods used to estimate LUC emissions are diverse. These include census-based historical land use reconstructions and land use statistics, satellite estimates of biomass change through time (Baccini et al., 2012), satellite monitored fire activity and burn area estimates associated with deforestation (van der Werf et al., 2010). In addition, there is increasing use of ecosystem models coupled with remote sensing to estimate emissions from LUC (Galford *et al.* 2011).

Emissions associated with the LUC sector have the highest degree of uncertainty given the complexity of processes involving net emissions and Houghton et al. (2012) assessed this uncertainty at $\sim 0.5$ Pg C $y^{-1}$, which is of the same order of magnitude as the emissions themselves. Uncertainties in estimating GHG emissions arising from savanna clearing, associated debris burning and conversion to agriculture are greater than those for tropical forests (Fearnside et al., 2009). It is important to quantify the emissions and their uncertainties in savannas particularly because tropical savanna woodland and grasslands occupy a large area globally (27.6 million $km^2$), greater than tropical forest (17.5 million $km^2$, Grace et al., 2006). Deforestation and associated fire from these biomes are the largest contributors to global LUC emissions (Le Quéré et al., 2014). Much of these GHG emissions are from the Brazilian Amazonia, an agricultural area that has been expanding

since the 1990s. However, over the last decade, the rate of tropical forest deforestation in this region

has decreased from 16,000 km$^2$ in early 2000s to ~6,500 km$^2$ by 2010 (Lapola et al., 2014), but at

the expense of the Brazilian cerrado, a vast savanna biome of some 2.04 million km$^2$, where

clearing rates have been maintained (Ferreira et al., 2013, 2016; Galford et al., 2013). Given the

suitability of the cerrado topography and soils for mechanized agriculture, the Cerrado may become

the principal region of LUC in Brazil (Lapola et al., 2014).

North Australia is one of the world's major tropical savanna regions, extending some 1.93

million km$^2$ across north-west Western Australia, the northern half of the Northern Territory and

Queensland (Fisher and Edwards, 2015). This biome occupies approximately one quarter of the

Australian continent and since European arrival, 5% has been cleared for improved pasture,

horticulture and cropping (Landsberg et al. 2011), making it one of least disturbed savanna regions

in the world (Woinarski et al., 2007). However, this small percentage equates to a substantial area

of 9.2 million hectares and LUC and associated economic development in northern Australia is a

government imperative and this is likely to involve expansion and intensification of grazing,

irrigated cropping, horticulture and forestry (Committee on Northern Australia, 2014). Drivers of

this potential expansion in food and fibre production include the exploitation of growing markets of

Asia as well as domestic factors such as the perception that land and water resources of north

Australia can provide a future agricultural resource base to offset the expected declines in

agricultural productivity in southern Australia due to adverse impacts of climate change (Steffan

and Hughes, 2013).

Historically, intensive agricultural developments in northern Australia have been implemented

based on limited scientific knowledge with dysfunctional policy and market settings, and as a result

there has been limited success (Cook, 2009). Future expansion needs to be underpinned by sound

understanding of the consequences of regional scale land transformation on carbon and water

budgets and GHG emissions. Any significant expansion in northern agricultural production would

require clearance of native savanna vegetation, with unknown increases in GHG emissions. Most
LUC studies occur at catchment, regional or biome scales (Houghton et al., 2012) and are not
underpinned by good understanding of underlying processes. However, there are an increasing
number of plot-scale studies using eddy covariance and chamber methods to provide direct
measures of net GHG fluxes from contrasting land uses (Lambin et al., 2013). These studies
typically compare microclimate and fluxes of GHGs from pastures and/or crops with adjacent forest
ecosystems under a range of management conditions (e.g. Anthoni et al. 2004; Zona et al. 2013) or
natural grasslands and different cropping types (e.g. Zenone et al., 2011). In tropical regions, there
is a focus on transitions from forest to pasture and from forest to crops for food or bioenergy
production (Galford et al., 2011;Wolf *et al.* 2011; Sakai *et al.* 2004).
There are few studies that directly measure GHG emissions and sinks prior to, during and after
LUC at a single site. Land use change can involve rapid changes in net GHG emissions over
varying temporal scales (minutes, hours, and seasonal cycles) and continuous flux measurements
are essential to capture the magnitude of these events (Hutley et al. 2005). However, there are no
direct observations of emissions from savanna clearing in northern Australia, contributing to the
uncertainty associated with the LUC sector in Australia's national GHG accounts (Commonwealth
of Australia, 2015a).
Our objective is to provide a comprehensive assessment of GHG emissions associated with
savanna clearing. Our aims are to 1) quantify the typical rates of $CO_2$ exchange of intact tropical
savanna and make comparative measurements from an analogue site that was to be cleared, 2)
quantify $CO_2$ fluxes before, during and after a clearing event, 3) estimate both $CO_2$ and non-$CO_2$
($CH_4$ and $N_2O$) GHG emissions arising from burning of cleared debris and 4) quantify ecosystem
scale GHG balance for this land use conversion and compare with emissions from savanna fire, a
significant source of GHG emissions across north Australia.
**2.0 Methods**

2        In this study we used a paired site approach, where concurrent fluxes of $CO_2$, water vapour and

energy were measured using eddy covariance towers from an uncleared savanna woodland site and
a similar savanna woodland site on the same soil type that was to be cleared, burnt and prepared for
agricultural production. Fluxes of $CO_2$ were monitored for 161 days prior to clearing at both sites
with observations continuing during the clearing event (deforestation) and for another 507 days
through phases of woody debris and grass curing, burning and soil preparation through raking and
ploughing. The entire observation period was 668 days. Flux observations of net $CO_2$ exchange
were combined with on-site biomass measurements and regionally calibrated pyrogenic emissions
factors to estimate emissions of $CO_2$, $CH_4$ and $N_2O$ (Meyers et al. 2012, Commonwealth of
Australia, 2015b) from burning of the cleared debris that was a key component of the land
conversion. Fire derived emissions were combined with net $CO_2$ fluxes from the land conversion
phases to provide a total net emission in units of $CO_2$-e for this LUC. In this paper, we use the term
deforestation to describe 'savanna clearing'. Deforestation is defined under Australia's National
Greenhouse Accounting system as the loss of forest/woodland cover due to direct human-induced
actions that fails to regenerate cover via natural regrowth or restoration planting (Commonwealth of
Australia, 2015a).
**2.1 Study sites**

19       Both savanna woodland sites were located within the Douglas-Daly River catchment

approximately 300 km south of Darwin, Northern Territory (Fig. 1). Both sites are OzFlux sites
(www.ozflux.org.au), with flux observations ongoing at the uncleared (UC) savanna site since 2007
(Beringer  et al. 2016; Beringer et al., 2011; Hutley et al., 2011). OzFlux is the regional Australian
and New Zealand flux tower network that aims to provide continental-scale monitoring of $CO_2$
fluxes and surface energy balance to assess trends and improve predictions of Australia's terrestrial
biosphere and climate (Beringer et al., 2016). The UC site is broadly representative of Australian
tropical savanna woodland found on deep, well drained sandy loam soils at sites with ~1000 mm
MAP (Table 1). The cleared savanna site (CS) was carefully selected to ensure the vegetation and
soils were as similar to the UC site as possible, and with topography suitable for eddy covariance
measurements.
Both sites were classified as savanna woodland (type 4B2, Aldrick and Robinson 1972,
1:50,000 mapping) with an overstorey cover of 30%, equivalent to the 'Eucalypt woodland' Major
Vegetation Group (MVG) of the National Vegetation Information System (NVIS, Commonwealth
of Australia, 2003). The sites were dominated by an overstorey of *Eucalyptus tetrodonta* (F.
Muell.), *Corymbia latifolia* (F. Muell.). Soils at both the UC and CS sites were red kandosols of the
haplic mesotrophic great group (Isbell, 2002), characterised as deep, sandy-loams (Table 1). The
long-term mean annual precipitation (MAP) ($\pm$ SD) at the UC site was estimated at 1180 $\pm$ 225 mm
(1970-2012, Australian Water Availability Project (AWAP), www.csiro.au/awap), similar to the CS
site at 1107 $\pm$ 342 mm (1985-2013, Bureau of Meteorology station, Tindal, NT). Slopes at both sites
were < 2% with a fetch of ~1.5 km at the UC site and ~1 km at the CS site. At both sites, 23 m
guyed masts were installed to support eddy covariance instruments at 21.5 m above-ground. The
tower at the CS site was moved three times to ensure adequate fetch was maintained according to
seasonal wind direction during clearing and phases of the land use conversion. Instrument height
was also adjusted given the height of the surface post-clearing and during the soil tillage phase
(Table 2).
Satellite-derived burnt area mapping is available across north Australia at 250 m resolution
(North Australian Fire Information system (NAFI), www.firenorth.org.au) and indicated that fires
had occurred within the flux footprint of the UC flux tower in 5 out of the last 13 years (2000-
2013), whereas no fires had occurred within the footprint of the CS site. The average fire return
time for the entire Australian savanna biome is 3.1 years (Beringer et al., 2015).
**2.2 Flux measurements and data processing**
Eddy covariance systems at both sites consisted of CSAT3 3-D ultrasonic anemometers
(Campbell Scientific Inc., Logan, USA) and a LI-7500 open-path $CO_2$ / $H_2O$ analysers (Licor Inc.,
Lincoln, USA). Flux variables were sampled at 10 Hz and covariances stored every 30 minutes. The
LI-7500 gas analysers were calibrated at approximately six month interval for the duration of the
data collection period and were highly stable. Mean daily rainfall, air temperature, relatively
humidity, soil heat flux ($F_g$, W m$^{-2}$) and volumetric soil moisture ($\theta_v$, m$^3$ m$^{-3}$) from surface to 2.5 m
depths were measured at both sites. The radiation balance was measured using a CNR4 net
radiometer ($F_n$, W m$^{-2}$) (Kipp and Zonen, Zurich).
Thirty minute covariances were stored using data loggers (CR3000, Campbell Scientific,
Logan) and data post processing and quality control was undertaken using the OzFluxQC system as
described by Isaac et al. (2016). In this system, data are processed through three levels; Level 1 is
the raw data as collected by the data logger, Level 2 are quality-controlled data and Level 3 are post
processed and corrected but not gap-filled data. Quality control measures at Level 2 include checks
for plausible value ranges, spike detection and removal, manual exclusion of date and time ranges
and diagnostic checks for all quantities involved in the calculations to correct the fluxes. Quality
checks make use of the diagnostic information provided by the sonic anemometer and the infra-red
gas analyser. Level 3 post processing includes 2-dimensional coordinate rotation, low- and high-
pass frequency correction, conversion of virtual heat flux to sensible heat flux ($F_h$, W m$^{-2}$) and
application of the WPL correction to the latent heat ($F_e$, W m$^{-2}$ and $CO_2$ fluxes ($F_c$) (Isaac et al.,
2016). Level 3 data also include the correction of the ground heat flux for storage in the layer above
the heat flux plates (Mayocchi and Bristow, 1995).
Gap filling of meteorology and fluxes along with flux partitioning of net ecosystem
exchange (NEE) into gross primary productivity (GPP) and ecosystem respiration ($R_e$) was
performed on the Level 3 data using the Dynamic INtegrated Gap filling and partitioning for Ozflux
(DINGO) system as described by Beringer et al., (2016b). In summary, DINGO gap fills
meteorological variables (air temperature, specific humidity, wind speed and barometric pressure)
using nearby Bureau of Meteorology (BoM, www.bom.gov.au) automatic weather stations that
were correlated with tower observations. All radiation streams were gap-filled using a combination
of MODIS albedo products (MOD09A1) and BoM gridded global solar radiation and gridded daily
meteorology from the Australian Water Availability Project (AWAP) data set (Jones et al. 2009).
Precipitation was gap-filled using either nearby BoM stations or AWAP data. Soil temperature and
moisture were filled using the BIOS2 land surface model (Haverd et al., 2013) run for each site and
forced with BoM or AWAP data. Energy balance closure was examined using standard plots of
$F_h + F_e$ vs $F_n - F_g$ using 30 minute flux data from both sites (data not shown). For the CS site, closure
was examined using data grouped according to the nine LUC phases as given in Table 2. For the
UC site, all 30 minute data from 2007-2015 was used.

12       Gap filling of fluxes was undertaken using DINGO that uses an Artificial Neural Network

(ANN) model following Beringer et al. (2007). Model training uses gradient information in a
truncated Newton algorithm. NEE and fluxes of sensible, latent and ground heat fluxes were
modelled using the ANN with incoming solar radiation, VPD, soil moisture content, soil
temperature, wind speed and MODIS EVI as inputs. The ustar threshold for each site was
determined following Reichstein *et al.* (2005) and night time observations below the ustar threshold
were replaced with ANN modelled values of $R_e$ using soil moisture content, soil temperature, air
temperature and MODIS EVI as inputs. The ANN $R_e$ model was then applied to daylight periods to
estimate daytime respiration and GPP was calculated as the difference between NEE and $R_e$. For
data collected at the CS site, a unique ANN model was developed for each LUC phase given the
differing canopy and microclimatology of each phase. At each site, daily NEE, $R_e$ and GPP were
calculated for each day of each phase.
**2.3 Leaf area index**
Canopy leaf area index (LAI) at the CS site in the surrounding intact savanna was measured
using a 180$^o$ hemispherical lens (Nikon 10.5 mm, f/2.8) after Macarlane et al. (2007). Three
savanna transects were photographed seasonally on 9 occasions over 2.1 years from the pre-clearing
phase (October 2011) to December 2013. Along each 100 m transect, 11 hemispherical pictures
were taken at 10 m intervals (33 photos for each measure occasion). At both sites the LAI was also
estimated using MODIS Collection 5 LAI (MOD15A2) for a 1 km pixel around each tower. The 8-
day product was interpolated to daily time series using a spline fit. Only MODIS values with a
quality flag of 0 for FparLai_QC were used in the estimate indicating the main algorithm was used
(lpdaac.usgs.gov/sites/default/files/public/modis/docs/MODIS-LAI-FPAR-User-Guide.pdf).
**2.4 Land use conversion**
The specific sequence and timing of clearing, burning and land preparation phases are given in
Table 2. Conversion of woodland to agricultural land in northern Australia is typically achieved by
pulling trees over using large chains held under tension between two bulldozers. Clearing occurs at
the end of the wet season when soil moisture is still high and soil strength low as under these
conditions trees are easily pulled over, with a large fraction of the tree root mass extracted when
pulled. At the CS site, 295 ha of savanna were deforested between 2 and 6 March 2012 using this
technique. A permit for this land conversion had been issued by the regional land management
agency following an impact assessment and erosion control planning. Chains were under tension
and intercepted tree boles 0.1- 0.2 m height above the ground which assisted in pulling the trees and
limited damage to the soil surface. As a result, grasses, woody re-sprouts and shrubs of the
understorey remained largely intact following deforestation (Plate 1a). Mechanised ripping of soil
to 60 cm depth was also undertaken to remove remaining coarse root material.
A cost-effective method of removing cleared vegetation is curing (drying) and subsequent
burning and the land managers at the CS site left debris onsite to for 5 months through the dry
season (March to August, 2011). Burning of debris occurred over a 22 day period in the late dry
season, August 2012 (Plate 1b), a period of consistent southerly trade winds of low relative
humidity (10-20%, BoM, Tindal station, NT). Prior to ignition, 100 m fire breaks were installed
around the entire 295 ha block and then lit in blocks of ~80 ha in size. There was an initial ignition
of the fine and coarse fuels (grasses, litter and twigs, defined below) and woody debris (heavy
fuels). Heavy fuels that were not completely consumed following the initial burn were then stock-
piled in rows ~1-2 m in height and re-ignited until the fuel was consumed (Plate 1c). Inspection of
debris post fire suggested ~5% of fine fuels remained as ash and ~10% of the heavy fuels remained
as charcoal, which were subsequently incorporated into the top soil on during soil bed preparation
(Plate 1d).
**2.5 GHG emissions from debris burning**
Emissions of $CO_2$, $CH_4$ and $N_2O$ from the debris burning were estimated following the
approach as outlined in the IPCC Good Practice Guidelines (IPCC 2003), which uses country or
region specific emission factors for fire activity (indicated by burnt area) and the mass of fuel
pyrolised to estimate the emission of each trace gas. This approach is well developed for the fire
regime of north Australian savanna and is described by Russell-Smith et al. (2013) and Murphy et
al. (2015a). These authors describe a novel GHG emissions abatement methodology for savannas
burning that combines indigenous fire practices with an emissions accounting framework, the
Emissions Abatement through Savanna Fire Management (Commonwealth of Australia 2015b,
www.comlaw.gov.au/Series/F2013L01165). This methodology is a legislative instrument that
establishes procedures for abatement projects for prescribed savanna burning and defines emission
factors for four fuel classes; fine (grass and litter < 6 mm diameter fragments), coarse (6 mm–5 cm),
heavy (>5 cm diameter) and shrubs fuels (Russell-Smith et al., 2013). Emissions of GHGs are
estimated based on vegetation type, fuel mass per area for each fuel type, burn area, the burning
efficiency (BEF) for each fuel type, defined as the mass of fuel exposed to fire that is pyrolised, the
fuel carbon content (%), elemental C:N ratios and emission factors (EF) for each GHG ($CO_2$, $CH_4$
and $N_2O$) and global warming potentials for each gas. Across north Australian savanna, values for
BEFs and EFs have been determined for both high (>1000 mm MAP) and low precipitation zones
(1000-600 mm MAP) and for both early and late dry season fires, which are fires occurring after 1
August which typically have higher intensity and combustion efficiencies than early dry season
fires (Russell-Smith et al. 2013).
We used these definitions of vegetation fuel type (woodland savanna with mixed grass) and
associated EF, carbon contents, N:C ratio values as defined in the methodology to estimate GHG
emissions from the debris fire using the following equation;
$$E = \sum_i (FL_j \times BEF_j \times CC_j \times N:C_j \times EF_{i,j} \times GWP_i) \qquad \text{Equation 1;}$$
where E is the sum of emissions in Mg $CO_2$–e ha$^{-1}$ for each GHG $i$ ($CO_2$, $CH_4$, and $N_2O$), $FL_j$ is the
fuel load for fuel type $j$ (fine, coarse, heavy) in Mg C ha$^{-1}$, $BEF_j$ is the burning efficiency factor, $CC_j$
is the fractional carbon content, $N:C_j$ is the fuel nitrogen to carbon ratio (for $N_2O$ emissions), $EF_{i,j}$ is
the emission factor for GHG $i$ and fuel type $j$ and $GWP_i$ is the global warming potential for each
GHG $i$ (after Commonwealth of Australia, 2015b). The debris fire differed from a typical savanna
fire in that there was a significantly higher heavy fuel load present and it was of high intensity
which consumed the vast majority of fuel (Plate 1c,d), reflected in the assumed BEFs we used. The
fire-derived emissions were combined with tower-derived NEE data from the post-clearing phases
(Table 3) to give a total emission in $CO_2$-e for this LUC.
**2.6 Quantifying fuel loads**
To accurately quantify emissions from the debris fire, fine, coarse and heavy fuels were
estimated using plots and transects established across the 295 ha deforestation area. For fine fuels,
six 100 m transects were randomly located and at 20 m intervals along each transect, all fine (grass,
woody litter) and coarse (twigs, sticks) fuels were harvested from 1 m$^2$ quadrats, dried and weighed
to give a mean fine and coarse fuel mass for the site. We assigned on-site coarse woody debris
(CWD), above-ground and below-ground biomass estimates to the heavy fuel class (>5 cm diameter
fragments). To quantify CWD, an additional six 100 m transects were randomly located across the
deforestation area and along each transect the length and diameter of all intersected CWD fragments
were recorded to estimate fragment volume. In these savannas, large fragments (>10 cm diameter)
are frequently hollowed from the action of termites and fire and the diameter and length of the
annulus of such fragments were measured to estimate this missing volume. In addition, large
fragments that were tapered were treated as a frustum of a cone and a second diameter was taken at
the fragment end to improve volume estimation. Fragment volumes were calculated and converted
to mass using rot classes (RC) and associated wood densities (g cm$^{-3}$). Five rot classes (RC) were
defined and assigned to each CWD fragment to capture the decay gradient of fragments. These were
defined as recently fallen, solid wood (RC1), solid wood with or without branches present but with
signs of aging (RC2), obvious signs of weathering, still solid wood, bark may or may not be present
(RC3), signs of decay with the wood sloughed and friable (RC4) and severely decayed fragments
with little structural integrity remaining (RC5). A wood density was assigned to each RC and
species (where identifiable) after Rose (2006) and Brown (1997) to provide an accurate estimate of
CWD mass that included decay and hollowing. For the dominant *Eucalyptus* and *Corymbia* species
wood densities ranged from 0.7 g cm$^{-3}$ (RC1) to 0.56 g cm$^{-3}$ (RC 5).
Above-ground biomass was quantified by surveying all woody plants >1.5 m in height or > 2
cm DBH across eight 50 x 50 m plots. All woody individuals were identified to species and stem
diameter at 1.3 m height (DBH) and tree height were measured. Region specific allometric
equations are available for tree species found at the CS site (Williams et al., 2005) and these were
used to estimate above-ground biomass for each individual tree and shrub based on DBH and
height. Below-ground biomass was calculated using the root:shoot ratio estimate of Eamus et al.
(2002) for these savanna stands which was 0.38. These trees have large lateral roots in the top 30
cm of soil, with no tap root and 90% of root biomass is found in the top 50 cm of soil (Eamus et al.
2002). As such, we assumed that chaining and bulldozer clearing of all above-ground biomass
followed by soil ripping (ploughing) to 60 cm soil depth, plus mechanised removal of root biomass
associated with tree boles and subsequent burning, resulted in a near-complete removal of both
above- and below-ground woody biomass pools (Plate 1d).
**2.7 Deforestation and savanna burning emissions at catchment to regional scales**

6        The potential impact of any expanded deforestation across north Australian savanna landscapes

was assessed relative to historic deforestation rates and resultant GHG emissions and arising from
prescribed savanna burning. This land management activity contributes ~3% to Australia's national
GHG emissions (Whitehead et al., 2014) and is 25% of the Northern Territory's annual emissions
(Commonwealth of Australia, 2015a). Annual emissions from these activities (historic and future
savanna deforestation and prescribed burning) were estimated at three spatial scales; 1) catchment,
2) state/territory and 3) regional. Emissions estimates from deforestation and savanna burning were
compiled for 1) the Douglas-Daly River catchment where the UC and CS sites are located (area
57,571 $km^2$), a catchment with less than 5% of the native vegetation deforested to date (Lawes et al.
2015) but earmarked for future development; 2) the savanna area of Northern Territory (856,000
$km^2$) and 3) the savanna region of north Australia as defined by Fox et al. (2001) with MAP > 600
mm, an area of 1.93 million $km^2$ (Fig. 1, insert).

18       Emissions of GHG from historic deforestation from the Douglas-Daly catchment were

estimated using our estimates for savanna land conversion combined with satellite-derived annual
deforestation area (1990-2013) as reported by Lawes et al. (2015) for this catchment to give a
catchment scale mean annual estimate of emissions from deforestation in Gg $CO_2$-e $y^{-1}$. Annual
deforestation emissions data for the Northern Territory and the north Australian savanna region
were taken from the National Greenhouse Gas Inventory (NGGI) for the same period 1990-2013.
The Department of Environment is responsible for reporting sources of greenhouse gas emissions
and removals by sinks in accordance with UNFCCC Reporting Guidelines on Annual Inventories
and the supplementary reporting requirements under the Kyoto Protocol. State and Territory GHG
Inventories are reported for 1990 to 2013 (Commonwealth of Australia, 2015a) and we used data
for the Land Use, Land-Use Change and Forestry sector, Activity A.2 Deforestation. These
emissions are reported for each State, but are not biome based and for our regional savanna
estimate, emissions data for Western Australia, the Northern Territory and Queensland were used
but were calculated using the area within each state that was defined as savanna by Fox et al. (2001,
Fig. 1). Mean annual deforestation emissions from the savanna area of each state and territory
(1990-2013) were summed to calculate a mean (±SD) annual deforestation rate for the north
Australian savanna area (1.92 million km$^2$) in Gg $CO_2$-e y$^{-1}$.
Emissions from savanna burning were calculated using the on-line Savanna Burning Abatement
Tool (SAVBat2, www.savbat2.net.au) using the pre-defined Vegetation Fuel Types (VFTs)
mapping for north Australian savanna (Fisher and Edwards, 2015; Thackway, 2014), both
components of the Emissions Abatement through Savanna Fire Management methodology.
SAVBat2 combines satellite derived burnt area mapping (www.firenorth.org.au) with fuel load
estimates from VFT mapping, GHG emission factors and burn efficiencies to estimate annual
emissions from burn areas. In accordance with IPCC accounting rules, only non-$CO_2$ emissions are
reported for savanna burning as it is assumed that $CO_2$ emissions from dry season burning is offset
by re-growth of vegetation (mostly $C_4$ grasses) in subsequent wet season(s) (IPCC, 1997).
However, for comparisons with deforestation emissions, we calculated emissions of $CO_2$ as well as
non-$CO_2$ emissions. SAVBat2 estimates were compiled for the same areas as savanna deforestation
estimates; the Douglas-Daly River catchment, savanna of the NT and north Australian savanna.
Mean annual burning emissions for 1990-2013 were calculated and are reported as non-$CO_2$ ($CH_4$,
$N_2O$) and total emissions ($CO_2$, $CH_4$ and $N_2O$) in Gg $CO_2$-e y$^{-1}$.
**2.8 Emissions from expanded deforestation across north Australia**
Emissions from expanded deforestation across north Australia was estimated by upscaling our
estimate of deforestation emissions per hectare from catchment areas identified as having future
clearing potential. These areas were based on the land use assessment of north Australian
catchments by Petheram et al. (2014) and identified catchments with development potential based
upon surface water storage and proximity of land resources suitable for irrigation development for
agriculture, horticulture or improved pastures. Using these criteria, suitable catchments were
identified in Western Australia (Fitzroy River, Ord Stage 3; 75 000 ha potential area), the Northern
Territory (Victoria, Roper Rivers, Ord Stage 3, Darwin-Wildman River area; 114, 500 ha) and
Queensland (Archer, Wenlock, Normanby, Mitchel Rivers; 120 000 ha). This gives a potential
savanna deforestation area of 311, 000 ha, equivalent to an additional 16% of cleared land over and
above the 1,886,512 ha that has been cleared across the savanna biome since 1990 (Commonwealth
of Australia, 2015a). Projected emissions included mean annual emissions from historic
deforestation rates plus emissions from this expanded deforestation scenario. Expanded
deforestation areas were calculated assuming any such clearing would occur over a five year period
and are reported as non-$CO_2$ ($CH_4$, $N_2O$) and total emissions ($CO_2$, $CH_4$ and $N_2O$) in Gg $CO_2$-e $y^{-1}$.
**3.0 Results**
**3.1 Pre-clearing site comparisons**
Pre-clearing meteorology, flux observations and energy balance closure for UC and CS sites
were compared (Fig. 2). Mean monthly NEE, $R_e$ and GPP for each LUC phase for both sites are
given in Table 3. Flux measurements prior to clearing were made for 161 days, a period spanning
the late dry to early wet season transition (September-December) through to the mid-wet season
(January-February, Table 2). Flux data at the CS site were validated by assessing energy balance
closure, with a regression between energy balance components suggesting closure was high with a
slope of 0.91 and an $R^2$ of 0.95 (n=4778). Site differences for each phase were tested using one-way
ANOVA using daily mean NEE with days as replicates. For Phase 1, mean daily NEE was not
significantly different between the two sites during (P<0.64, df=321). Seasonal patterns of $T_{air}$,
VPD (Fig. 2b), LAI (Fig 2c) and C fluxes (NEE, GPP, $R_e$, Fig 2d) were similar when both sites
were intact, although precipitation was 340 mm higher at the UC site (Table 3).

4        At both sites, NEE shifted from being a weak sink of less than -1 μmol $CO_2$ m$^{-2}$ s$^{-1}$ during the

late dry season to a net source of $CO_2$ during the early wet season (Fig. 2d). During this period, $R_e$
increased rapidly from +2 μmol m$^2$ s$^{-1}$ to +5 μmol m$^2$ s$^{-1}$ in early October with the onset of wet
season rain, but then remained relatively constant for the remainder of the wet season. As the wet
season progressed, temporal patterns of GPP were similar at both sites and steadily increased to -6
to -7 μmol m$^{-2}$ s$^{-1}$ and remained at this rate until cleared (March 2012). Re was relatively stable
during this period and NEE increased to -2 μmol m$^{-2}$ s$^{-1}$ through the wet season (December to
February). Despite the higher precipitation received at the UC site, mean monthly NEE, GPP and $R_e$
differed by <10% (Table 3, intact canopy phase). Normalising fluxes by MODIS LAI for each site
further reduced differences to 2% (data not shown), suggesting site differences were small and the
UC site provides a suitable control for the CS site.
**3.2 Fluxes following clearing**

16       Clearing of the 295 ha block commenced on 2 March 2012 and the bulldozers reached the

footprint of the flux tower at ~0900h local time on 6 March (Fig. 3). As for Phase 1, energy balance
closure of flux tower data for LUC Phases 2 to 4 (post-clearing phases) was high, with a slope >0.9
and $R^2$ > 0.92. Over all phases at the CS site, closure was lower, with a slope of 0.81 ($R^2$ =0.95,
n=26,395), similar to that of the UC site at 0.87 ($R^2$ =0.93, n=99,998).

21       The four day clearing event occurred during relatively high soil moisture conditions, with

surface (5 cm depth) $θ_v$ ranging from 0.08 to 0.10 m$^3$ m$^{-3}$ and sub-soil $θ_v$ (50 cm depth) ranging
from 0.12 to 0.14 m$^3$ m$^{-3}$. As a result, pre-clearing fluxes were high and NEE reached -15 μmol $CO_2$
m$^{-2}$ s$^{-1}$ during the middle of the day (Fig. 3). Mean daily NEE for the week prior to clearing was a
net $CO_2$ sink of -0.60 ± 0.63 μmol m$^{-2}$ s$^{-1}$, and was not significantly different to mean daily NEE at
the UC site of -0.80 ± 0.93 µmol m$^{-2}$ s$^{-1}$ (ANOVA, P<0.03). For the three weeks following clearing,
the CS site rapidly became a net source of $CO_2$ with a mean daily NEE of +4.38 ± 0.24 µmol m$^{-2}$ s$^{-}$
$^{1}$, with a much reduced diurnal amplitude and no response to precipitation events (Fig 3a,b). High
closure (slope>0.9) was observed during Phases 2 to 4, although this was reduced (slope=0.75) for
the post-fire and soil preparation, Phases 6-9.
Table 3 provides values of precipitation and monthly NEE, Re and GPP for the seven LUC
phases following clearing, namely debris decomposition and curing (153 days), burning (22 days),
wet season regrowth (80 days), followed by soil tillage and preparation of irrigated raised soil beds
(181 days). For each phase, the comparable flux estimate from the UC site is estimated for all post
clearing phases and for the entire observation period. Following clearing, GPP at the CS site was
reduced by a factor of 3.5 when compared to the UC for the same period (March 2012 – January
2013, Table 3). While greatly reduced, GPP still occurred at the CS site during this 13.7 month
period (-0.38 Mg C ha$^{-1}$ month$^{-1}$), via re-sprouting of felled overstorey and sub-dominant trees and
shrubs, as well as grass germination and growth stimulated by early wet season precipitation
(November 2012-January 2013, 361 mm, Table 3). Ecosystem respiration during this period was
higher at the UC site (+1.12 Mg C ha$^{-1}$ month$^{-1}$) when compared to the CS site (+0.82 Mg C ha$^{-1}$
month$^{-1}$) and given the large decline in GPP, the CS site was a small net C source at +0.51 Mg C ha$^{-}$
$^{1}$ month$^{-1}$, as compared to the UC site which was a weak sink of -0.03 Mg C ha$^{-1}$ month$^{-1}$.
Cumulative NEE over all the post-clearing LUC phases was +7.2 Mg C ha$^{-1}$ at the CS site as
compared to a net sink of -0.78 Mg C ha$^{-1}$ at the UC site (Table 3). The temporal dynamics of
cumulative NEE across all LUC phases (note differences in phase duration) is summarised in Fig. 4,
which compares fluxes from both sites for the complete observation period. Three significant
periods of C emission are evident in Fig. 4. Firstly, the clearing event and the subsequent switch
from a C sink to a net source of 1.9 Mg C ha$^{-1}$ due to soil disturbance and the decomposition of
biomass. Secondly, this was followed by a reduction in source strength over the dry season of 2012,
attributable to a reduction in $R_e$ during the dry season (2012 dry season pre-burn phase, Table 3).
Thirdly, there were other major emissions attributed to soil tillage and bed preparation in the wet
and dry seasons of 2013, a cumulative net emission of +2.75 Mg C ha$^{-1}$ that occurred over the final
six months (Fig. 4) in preparation for cropping. Over this phase, the UC site was a net sink of -0.62
Mg C ha$^{-1}$.
**3.4 Emissions from debris burning**
Table 4 gives fuels loads, BEF, EF, carbon content and N:C ratios for each fuel type used to
estimate the GHG emission from the debris burning. Fuel load was dominated by heavy fuels with a
mean (±SD) above-ground biomass of 26.9 ± 7.0 Mg C ha$^{-1}$ and a range of 14.4 to 39.3 Mg C ha$^{-1}$
across the eight biomass plots. The mean below-ground biomass was estimated at 9.0 ± 2.4 Mg C
ha$^{-1}$ and CWD was 1.4 ± 0.6 Mg C ha$^{-1}$. Fine and coarse fuels were 1.4 ± 0.7 and 0.5 ± 1.0 Mg C ha$^{-}$
$^{1}$ respectively, giving a total fuel mass of 38.2 Mg C ha$^{-1}$. Using these fuel loads with savanna EF
and the BEFs estimated for the site gave an emissions of $CO_2$, $CH_4$ and $N_2O$ for each fuel type and
the emission from debris burning totalled 121.9 Mg $CO_2$-e ha$^{-1}$, with 9.5% of this total comprising
non-$CO_2$ emissions (Table 4).
**3.5 Total GHG emission**
Emissions derived from debris burning needs to be combined with the post-clearing NEE as
measured by the EC system to provide a total GHG emissions estimate from this LUC in units of
$CO_2$-e. The LUC phases following clearing spanned a 502 day period (Table 3), and NEE was +7.2
Mg C ha$^{-1}$ or +26.4 Mg $CO_2$-e ha$^{-1}$. In comparison, NEE from the UC site over the same period was
-0.78 Mg C ha$^{-1}$ or -2.9 $CO_2$-e ha$^{-1}$. Adding NEE from post-clearing phases (Phases 2-9, Table 3) to
emissions from debris burning (Table 4) gave a total emission of +148.3 Mg $CO_2$-e ha$^{-1}$ for the CS
site. The $CO_2$-only emission from debris burning plus post-clearing NEE was +136.7 Mg $CO_2$ ha$^{-1}$,
which was a flux 45 times larger than the observed savanna $CO_2$ sink at the UC site over the post-
clearing period.
**3.6 Upscaled and projected emissions from deforestation and savanna burning**
Table 5 provides mean (±SD) GHG emissions estimates for savanna burning and deforestation
for 1990-2013. At all spatial scales, annual mean burnt area dwarfed the mean annual land area
deforested. For the Douglas-Daly catchment area, reportable non-$CO_2$ emissions from savanna
burning were $577\pm124$ Gg $CO_2$-e $y^{-1}$, almost four times larger than emissions from the mean annual
savanna deforestation rate of $163\pm162$ Gg $CO_2$-e $y^{-1}$. For the Northern Territory savanna, mean
annual burning emissions were an order of magnitude larger than mean annual deforestation
emissions (Table 4) and two orders of magnitude larger if $CO_2$ emissions were included. At a
regional scale, the mean annual deforestation rate across the north Australian savanna was
$16,161\pm5,601$ Gg $CO_2$ $y^{-1}$, with emissions from Queensland savanna area dominating this amount at
$15,762\pm5,566$ Gg $CO_2$ $y^{-1}$. This is double that of the reportable (non-$CO_2$ only) emission from
prescribed burning at $6,740\pm1,740$ Gg $CO_2$ $y^{-1}$ (Table 5).
Emissions estimates that include future deforestation rates would be equivalent to savanna
burning, at least for the duration of the additional deforestation. For the Douglas-Daly catchment,
this future emission is estimated at 756 Gg $CO_2$-e $y^{-1}$ and across the Northern Territory savanna
area, this would be 3,413 Gg $CO_2$-e $y^{-1}$, rates of emission that are equivalent to burning emissions
catchment (Douglas-Daly, $577\pm124$) and state scales (Northern Territory savanna, $3,490\pm922$ Gg
$CO_2$-e $y^{-1}$). Emissions that include future deforestation rates for the north Australian savanna region
were estimated at 24,393 Gg $CO_2$-e $y^{-1}$ and would be three times the reportable savanna burning
annual emissions (Table 5).
**4.0 Discussion**

1   Australia has lost approximately 40% of its native forest and woodland since colonisation

2 (Bradshaw, 2012), with most of this clearing for primary production in the eastern and south-eastern

3 coastal region. Attention has now turned to the productivity potential of the largely intact northern

4 savanna landscapes, which will involve trade-offs between management of land and water resources

5 for primary production and biodiversity conservation (Adams and Pressey, 2014; Grundy et al.,

6 2016). Globally and in Australia, savanna fire ecology and fire derived GHG emissions have been

7 reasonably well researched (Beringer et al., 1995; Cook and Meyer, 2009; Livesley et al., 2011;

8 Meyer et al., 2012; Walsh et al., 2014; van der Werf et al., 2010) and the impacts of fire on the

9 functional ecology of Australian savanna has been recently reviewed by Beringer et al. (2015). In

10 this study, we focussed on savanna deforestation and land preparation for agricultural use. These

11 phases result in a series of events that may lead to pulsed GHG emissions that would otherwise be

12 missed or greatly under-estimated by episodic measurements taken at a weekly or monthly

13 frequency after an initial tree felling event (Neill et al., 2006; Weitz et al., 1998).

14   We used the eddy covariance methodology as it provides a direct and non-destructive

15 measurement of the net exchange of $CO_2$ and other GHG gases at high temporal resolution, ranging

16 from 30 minute intervals to daily, monthly, seasonal and annual estimates. The method is useful as

17 a full carbon accounting tool as all exchanges of $CO_2$ from autotropic and heterotrophic components

18 of the ecosystem undergoing change are quantified (Hutley et al., 2005). This approach provides

19 essential data for bottom-up GHG and carbon accounting studies as micrometeorological conditions

20 and associated fluxes can be tracked through time for the duration of a land use conversion.

21   At the CS site, burning of post-clearing debris of comprised 82% of the total emission of 148.4

22 Mg $CO_2$-e ha$^{-1}$, with the remainder attributed to NEE as measured by the flux tower. This flux

23 comprised significant $CO_2$ losses via respiration of debris, enhanced soil $CO_2$ efflux from soil

24 disturbance and tillage, which was partially offset by net uptake of $CO_2$ from woody re-sprouting

25 post-clearing and periods of grass growth following wet season rainfall (Fig. 4). Soil disturbance via

ripping, tillage and preparation was responsible for 10% of the $CO_2$ emission from the conversion.
The EC flux tower was operational during the clearing event, demonstrating the utility of this
method as the switch of the ecosystem from being a net $CO_2$ sink to being a net source occurred
over a number of hours as the clearing event was completed (Fig. 3). During the LUC phase
changes, there was little evidence of major pulses of $CO_2$ flux, instead there was a rapid transition
to a new diurnal pattern following the clearing (Fig. 3) or the commencement of soil preparation
(data not shown). This is in contrast to non-$CO_2$ flux emissions, in particular $N_2O$, with short term
emissions often follow disturbance (Grover et al., 2012; Zona et al., 2013) and can be a significant
fraction of annual emissions.
The net $CO_2$ source as measured by the flux tower represents an emission that would be missed
if vegetation biomass density alone was used to estimate LUC emissions, the approached used in
current remote sensing LUC studies at regional and continental scales, data that is the basis of
emissions reporting for the LULUC sector. The total GHG emission we report in this study is more
accurately described as a land conversion, as it includes the oxidation of biomass plus emissions
associated with soil disturbance and tillage required for a conversion to a cropping or grazing
system.
The emission estimate from this study does not include non-$CO_2$ soil derived fluxes of $CH_4$ and
$N_2O$, which can be significant for LUC events in certain ecosystems (Tian et al., 2015). Grover et
al. (2012) compared soil $CO_2$ and non-$CO_2$ fluxes from native savanna with young pasture and old
pastures (5-7 and 25-30 years old) in the Douglas-Daly River catchment. Soil emissions of $CO_2$-e
were 30% greater on the pasture sites as compared with native savanna sites, with this change being
dominated by increases in $CO_2$ emission and soil $CH_4$ exchange shifting from a small net sink to a
small net source at the pasture sites. Non-$CO_2$ soil fluxes were generally small, especially $N_2O$
emissions, although these measurements were made many years after the LUC event and there is
uncertainty as to their relevance for a recently deforested and converted savanna site. An additional
pathway for $CH_4$ and $N_2O$ emissions in these savannas is via termite activity (Jamali et al., 2011a,
2011b). In our study, termite mounds were abundant across the CS site but were largely destroyed
by clearing and soil preparation, potentially reducing the net non-$CO_2$ emission following
conversion. Further work is required to quantify these non-$CO_2$ fluxes not associated with debris
burning to refine our total emission estimate for savanna deforestation.
This land conversion represents the loss of decades of carbon accumulation in this mesic
savanna (>1000 mm MAP), ecosystems which are currently thought to be a weak carbon sink
(Beringer et al., 2015). The 8-year ensemble mean NEE for the UC site was -0.11 ± 0.16 Mg C ha$^{-1}$
y$^{-1}$ and is representative of a savanna site at a near-equilibrium state in terms of carbon balance
given the low fire frequency (3 in 13 years, Table 1) with high severity fires uncommon (1 in 8
years of flux measurements). The annual increase in tree biomass at this UC site is 0.6 t C ha$^{-1}$ y$^{-1}$
(Rudge, Hutley, Beringer, unpublished data), and given an above-ground standing biomass of 28 t
C ha$^{-1}$ suggests a regeneration period of approximately four to five decades after stand replacement
disturbance event such as deforestation for this savanna type.
Even after the large pool of carbon is lost following oxidation of biomass, carbon loss may
continue on cleared land via continued soil carbon mineralisation, leading to a slow decline in soil
carbon storage that is frequently reported for forest to cropping LUC (Jarecki and Lal, 2003; Lal
and Follett, 2009). Conversion of forest or woodland to improved pasture grazing may result in
either increases or decreases in soil carbon (Sanderman et al., 2010). Alternatively, it is possible
that carbon sequestration may occur post-clearing via woody regrowth if a cleared site is abandoned
and not further prepared for cultivation. This has actually been a relatively common transition and a
significant sequestration pathway that needs to be included in savanna LUC assessments (Henry et
al. 2015). Admittedly, if savanna cleared land does fully transition to a cropping system, some
fraction of the lost carbon could also be replaced or sequestered by new horticultural or forestry
land uses.
There are few detailed, plot scale studies of GHG emissions from savanna clearing in north
Australia. Several studies (Law and Garnett 2009, 2011) used the Full Carbon Accounting Model
(FullCAM Ver 3.0, Commonwealth of Australia, 2015a; Richards and Evans, 2004) to generate
spatial maps of above- and below-ground biomass and soil organic carbon pools across the NT. The
FullCAM model uses spatial and temporal soil, climate, precipitation data with NVIS major
vegetation classes to simulate carbon losses (as GHG emissions) and uptake between the terrestrial
biological system and the atmosphere. Land use change scenarios can be run within the model and
Law and Garnett (2009) examined deforestation emissions from the Eucalypt woodland NVIS
vegetation class, as per UC and CS site classification. Modelled emissions were $136\pm42$ Mg $CO_2$-e,
comparable to our deforestation estimate of 121.4 Mg $CO_2$-e. Henry et al. (2015) used a life cycle
assessment approach to quantify GHG emissions from LUC associated with beef production in
eastern Australia. Australia's major beef producing areas across central and southern Queensland
and northern central New South Wales were classified into 11 bioregions, with the northern most
bioregion, the northern Brigalow Belt, falling within the savanna biome. Vegetation biomass from
this bioregion was estimated at $84.7\pm7.1$ Mg $ha^{-1}$ or $\sim41.4$ Mg C $ha^{-1}$, with an emission estimated at
129 Mg $CO_2$-e (Henry et al., 2015), similar to the woodland biomass density and resultant emission
with deforestation from the CS site of this study.
Our emissions estimate is robust for this vegetation class and can be upscaled and compared
with other land sector activities such as prescribed savanna burning. At a regional scale, current
levels of savanna burning dominate emissions compared to land clearing rates (Table 5). The
cumulative deforestation area across the savanna region since 1990 (1,886,512 ha) is 17 times
smaller than the mean annual savanna burn area (32 Mha, Table 5) as approximately 30 to 70% of
the savanna area is burnt annually (Russell-Smith et al., 2009). Modelling NEP for savanna biome
for 1990-2010 (Beringer et al., 2015; Haverd et al., 2013) suggests the north Australian savanna is
near carbon neutrality, or is a weak source of $CO_2$ to the atmosphere once regional scale fire
emissions are included. As such, the IPCC assumption that $CO_2$ emissions from the previous year's
burning are recovered by the following year's wet season growth may have some validity for
regional scale GHG accounting. This assumption at plot to catchment scales may not be valid, as
localised interannual variability in rainfall, site history and fire management can result in either net
accumulation or loss of carbon (Hutley and Beringer, 2011; Murphy et al., 2014, 2015b). Assuming
year to year $CO_2$ emitted from burning is re-sequestered, assessment of the non-$CO_2$ only emissions
from savanna burning with deforestation is useful. This comparison suggests projected deforestation
emissions (24,393 Gg $CO_2$-e $y^{-1}$, Table 5) could be well in excess of current annual burning
emissions (6,740 Gg $CO_2$-e $y^{-1}$, Table 5), at least for the period of enhanced clearing, which in this
study we assumed to be five years.

10       In 2013, Australia's total reported GHG emission was 548,440 Gg $CO_2$-e and the impact of

expanded savanna deforestation on the national emission can be estimated using data in Table 5
which provide estimates of mean annual emissions from deforestation area, giving a mean annual
deforestation emission per ha averaged for the entire savanna area, which is 221 ±50.8 Mg $CO_2$-e
$ha^{-1}$ using 1990 to 2013 data (Commonwealth of Australia, 2015a). This value represents a spatially
averaged emission as it is derived from the full range of savanna vegetation types and above-ground
biomass, which across the Northern Territory savanna area ranges from 10 to 70 Mg C $ha^{-1}$ (Law
and Garnett, 2011). Assuming this emission per ha, an additional 311, 000 ha of savanna
deforestation, cleared over a five year period, adds 12,099 Gg $CO_2$-e $y^{-1}$. For the duration of the
expanded deforestation, this is a 2.2% increase to Australia's nation emission over and above the
historic savanna LUC emissions (16,161 Gg $CO_2$-e $y^{-1}$), which are 2.9% of national emissions.
Using our finding from flux tower measurements that a land conversion (deforestation followed by
site tillage and preparation for cultivation) adds an additional 18% of GHG emissions to a
deforestation event, expansion of northern land development could add an additional 3% or 33, 350
Gg $CO_2$-e $y^{-1}$ to the reportable national GHG emissions for the duration of the expanded
deforestation period.
This assessment is subject to a number of uncertainties. Firstly, a component of our emissions
estimate is based on eddy covariance measurements of $CO_2$ flux, which typically have an error of
10-20% (Aubinet et al., 2012). In this study, energy balance closure suggested fluxes were
underestimated by up to 13% across the entire observation period. Energy balance closure ranged
from <10% flux loss during the intact canopy phase to >20% error during the final three LUC
phases when the flux instruments were at 3 m height measuring net soil $CO_2$ emissions from the
smoothed, vegetation-free ploughed soil surface during preparation. Secondly, it is difficult to
predict the nature of future deforestation (rate, area, specific location) and the emission comparisons
presented here are indicative only. Catchments selected by Petheram et al. (2014) regarded as
suitable or with potential for future development were based on biophysical properties only, were
unconstrained by the regulatory environment and did not account for conservation and cultural
values placed on identified land and water resources. In addition, challenges to agricultural
expansion in northern Australia include uncertain land and water tenure, high development costs
and lack of existing water infrastructure, logistics and technical constraints, lack of human capital
and distance to markets, all factors that may restrict land clearing. It is well understood that the
availability and cost of water for irrigated, or irrigation assisted agriculture is critical for viable
agriculture in northern Australia (Petheram et al., 2008, 2009). Australian Government policies
currently support small-scale, precinct or project scale approaches, based on well-understood water
and soil resources, where water allocation is capped. The current policy and market instruments are
likely to ensure that development remains measured and restricted, unlike development of previous
decades in other regions of eastern and southern Australia.
As a result we used a conservative estimate of potential land suitability area (311, 000 ha over a
five year clearing period), as estimates of assumed clearable area ranging up to 700,000 ha (e.g.,
Douglas-Daly catchment, Adams and Pressey, 2014) or over 1 million ha across north Australia
(Petheram et al. 2014), areas that may be unlikely given capital investment requirements as well as
conservation and cultural considerations. Our comparison with burning emissions is also influenced

1 by the deforestation period we assume. This was based on patterns of historic rates of clearing as

2 there are periods when deforestation rates have easily exceeded 311,000 ha over five year periods,

3 particularly in Queensland (Commonwealth of Australia, 2015a) and a longer duration of

4 deforestation reduces the impact on annual national GHG accounting.

5  There is also uncertainty arising from our emissions from debris burning. Russell-Smith et al.

6 (2009) estimated errors associated with emissions estimates from the Western Arnhem Land Fire

7 Abatement (WALFA) project, a savanna burning based GHG abatement scheme operating in the

8 Northern Territory. This is a project area of the 23,893 $km^2$ consisting of a wide range vegetation

9 types including open-forest and woodland savanna and sandstone heaths in escarpment areas.

10 Russell-Smith et al. (2009) estimated the accountable emissions from savanna burning at 272 ±100

11 Gg CO2-e $y^{-1}$ (95% CI), an error of 30–35% of the mean. Uncertainty was ascribed to errors in

12 remotely sensed burn area mapping, fuel load estimation, spatial variation of fire severity, errors in

13 BEF for each fuel class and EFs. At the spatial scale of our study area, there were no uncertainties

14 with the burnt area, vegetation structure or fuel type classification, and we used site-specific fuel

15 load estimations used in our calculations, all of which would reduce the error associated with our

16 fire emissions estimate. Russell-Smith et al. (2009) also reported low coefficients of variability

17 (CV%) of for BEFs across fine, course and heavy fuel types for high severity fires, ranging from 0.3

18 to 11% and 2% CV for EFs for $CH_4$ and $N_2O$. Site specific sources of error include high spatial

19 variability of on-site fuel loads which had a CV% of ~70% (Table 4) and uncertainty associated

20 with the BEF we assumed for coarse and heavy fuel loads (0.9), which is higher than that derived

21 for late dry season savanna fires (0.36, 0.31 respectively, Russell-Smith et al. 2009). This value was

22 assumed as repeat burning of coarse and heavy fuels ensured ~10% of biomass remained as ash and

23 charcoal at the CS site. This assumed BEF is also consistent with FullCAM (4.00.3) BEF of 0.98

24 for forest fire with 100% of trees killed, although this is setting is based on Surawski et al. (2012)

25 who found little empirical evidence for BEF for stand-replacement fires. However, given the

detailed on-site measurements of fuel load, error in our fire-derived emissions would be of the order
of 20% or less.
**5.0 Conclusions**
While GHG emissions from savanna deforestation are dominated by debris burning, emissions
from soil tillage and soil bed preparation is likely to be 20% of the total emission, suggesting
satellite-based emissions based on oxidation of cleared vegetation alone do not capture all phases of
LUC prior to cultivation. Savanna burning, using the area as defined in this study, was 1.5% of
Australia's national GHG emissions and is of similar magnitude to emissions associated with
historic savanna deforestation. However, for the deforestation scenario could increase Australia's
GHG emissions by at least 3% per annum for the duration of the expansion, depending on the area
and deforestation rate. These are indicative estimates only, but suggest that the impacts of northern
agricultural development will have an impact on the national GHG budget and will need to be
considered in northern land use decision making processes. These considerations are also
particularly relevant given the emission reduction targets set by Australia following the 21st
Conference of Parties to the UN Framework Convention on Climate Change (COP21 / CMP11) to
reduce GHG emissions by 26 to 28% of 2005 by 2030.

# Acknowledgements

Financial support for this study was provided by the Australian Research Council's Linkage Project LP100100073 and Discovery Project DP0772981. Beringer is funded under the Australian Research Council's Future Fellowship program (FT1110602). Support for flux data collection and archiving was provided by Dr Peter Isaac of the Australian flux network, OzFlux (www.ozflux.org.au), which is funded by the Australian Terrestrial Ecosystem Research Network (TERN, www.tern.org.au). Chris and Bridget Schulz provided access to the property and field assistance throughout all phases of the land use change we monitored. We are grateful for the technical expertise and field assistance of Matthew Northwood and Michael Brand who maintained the eddy covariance tower. Yan-Shih Lin, Amanda Lilleyman and Allison O'Keefe provided field support during the intensive field campaigns. Thanks also to the Department of Environment for provision of savanna specific deforestation GHG emissions data, 1990-2013 and the two reviewers who provided constructive comments on the original manuscript.

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

**Table captions**
Table 1 Site characteristics for the uncleared savanna (UC) and cleared (CS) sites. Site soil orders
are given as per Isbell (2002) with savanna vegetation classified using Fox et al. (2001). Fire
frequency was estimated from fire mapping taken from the North Australian Fire Information
system (NAFI, www.firenorth.org.au) for 2000-2012. The fire frequency estimate for the CS site
excluded the debris fires in August 2012. Basal area and stem density is provided for all woody
stems >2 cm DBH at both sites. Mean site LAI for the UC is taken from Hutley et al. 2011 and for
the CS site, was estimated from canopy hemispherical photos, see text for details.
Table 2 Characteristics of land conversion phases during the 668 day observation period at the
savanna clearing site (CS). Also given are the canopy heights following LUC phases and flux
instrument heights that were adjusted following clearing, burning and then soil preparation phases.
Table 3 Cumulative precipitation and mean NEE, Re and GPP (Mg C ha$^{-1}$ month$^{-1}$) for each of the
LUC phases at the CS site as measured by the flux tower. These fluxes are given for the UC site for
these same periods. One-way ANOVA was used to test for differences between mean daily NEE for
each LUC phase with significantly different means labelled with an asterisk. On the days of ignition
during the debris burning phase, flux data at the CS site were excluded. Integrated fluxes are given
for the post-clearing period (507 days) and the entire observation period (668 days) for both sites in
Mg C ha$^{-1}$.
Table 4 Measured fuel loads, assumed burning efficiencies (BEF), carbon contents, N:C ratio and
emissions factors (EF) used to estimate GHG emissions from the burning of the post-deforestation
fine, coarse and heavy fuel debris. Emission factors, carbon content and C:N ratio were assumed for
the vegetation fuel type woodland savanna with mixed grass (code hWMi) as given in the
Emissions Abatement through Savanna Fire Management methodology (Commonwealth of
Australia, 2015b), available at www.legislation.gov.au/Details/F2015L00344 and Meyers et al.

26    2012.

Table 5 Greenhouse gas emissions for 1990-2013 from prescribed savanna burning and savanna
deforestation at catchment (Douglas-Daly River), state/territory (Northern Territory savanna area)
and regional scales (north Australian savanna area, Fig. 1). For savanna burning, burnt area and
associated mean annual emissions (± SD) are given for both reportable non-CO2 (CH4, N2O) and
total emissions ($CO_2$, $CH_4$ and $N_2O$). For the identical areas as used for savanna burning, mean
annual GHG emissions from deforestation (± SD) are given. For the Douglas-Daly River
catchment, deforestation area was taken from Lawes et al. (2015) and combined with deforestation
emissions from the CS site. Deforestation emissions (1990-2013) for the NT and the north
Australian savanna area are taken from the State and Territory Greenhouse Gas Inventories
(Commonwealth of Australia, 2015a). In bold text are the emissions associated with the current
deforestation rate plus expanded deforestation areas as identified by Petheram et al. (2014), which
are combined with emissions from the CS site to give an upscaled estimate of potential emissions
with agricultural development at the three spatial scales.
Table 1

| Site | UC | CS |
|------|-----|-----|
| Location | 14°09'33.12"S, 131°23'17.16"E | 14°33'48.71"S, 132°28'39.47"E |
| Soils | Red Kandosol | Red Kandosol |
| Vegetation type | Savanna woodland with mixed grasses<br><br>Map unit **D4**. *E. tetrodonta, C. latifolia, Terminalia grandiflora, Sorghum spp, Heteropogon triticeus* | Savanna woodland with mixed grasses<br><br>Map unit **D4**. *E. tetrodonta, Erythrophleum chlorostachys, Corymbia. bleeseri, Sorghum spp, H. triticeus* |
| Map unit area ($km^2$) | 59,986 | 59,986 |
| Fire frequency ($y^{-1}$) | 0.23 | 0.07 |
| Basal area ($m^2$ $ha^{-1}$) | 8.3 | 6.8 |
| Canopy height (m) | 16.4 | 14.2 |
| Above-ground biomass (Mg C $ha^{-1}$) | 30.6 ± 9.2 | 26.2 ± 7.0 |
| Stem density ($ha^{-1}$) | 330 ± 58 | 643 ± 102 |
| Overstorey LAI (wet/dry) | n/a / 0.8 | 0.9 / 0.5 |
| MODIS LAI (wet/dry) | 1.5 / 0.9 | 1.6 / 1.0 |
| MAP (mm) | 1372[a] /1180[b] | 1107[c] |
| Max $T_{air}$ (°C) | 37.5 (Oct) / 31.2 (Jun) | 37.5 (Oct) / 29.7 (Jun) |
| Min $T_{air}$ (°C) | 23.8 (Jan) / 12.6 (Jul) | 25.0 (Nov) / 13.7 (Jul) |

[a]On-site observations, 2007-2012, [b]gridded precipitation (AWAP, 1970-2012), [c]Tindal BoM station (14.52S, 132.38E, data from 1985-2013).

Table 2

| Season | Period | LULUC phases | Canopy height (m) | Instrument height (m) |
|---|---|---|---|---|
| Late dry season | Sep - Oct 2011 | Intact savanna | 16 | 21.5 |
| Wet season pre-clearing | Oct 2011 - Feb 2012 | Intact savanna | 16 | 21.5 |
| Wet season clearing | Mar - May 2012 | Savanna deforested using bulldozers, followed by debris decomposition, understory grass germination | 3 | 7 |
| Dry season pre- burn | May - Aug 2012 | Vegetation debris curing, understorey grass growth | 2 | 7 |
| Debris burning | Aug 2012 | Debris and grasses burnt, soil ripped to 60 cm to remove roots, roots and remaining debris stockpiled, re-burnt | 2 | 7 |
| Dry season post-burn | Aug - Nov 2012 | Grass and shrubs germination and resprouting | 1 | 7 |
| Early wet season | Nov 2012 - Jan 2013 | Removal remaining below-ground biomass. Wet season rains stimulates grass growth, shrub re-sprouting and growth | 1 | 7 |
| Wet season | Jan - Mar 2013 | All regenerated vegetation removed, soil bed preparation | 0 | 3 |
| Dry season | Apr - Jul 2013 | Soil cultivation in stages | 0 | 3 |

Table 3

| LULUC phases | Phase number | Period (d) | CS Rainfall (mm) | NEE (Mg C ha$^{-1}$ month$^{-1}$) | Re | GPP | UC Rainfall (mm) | NEE (Mg C ha$^{-1}$ month$^{-1}$) | Re | GPP |
|---|---|---|---|---|---|---|---|---|---|---|
| Intact canopy cover | 1 | 161 | 736.6 | -0.23 | 1.57 | -1.79 | 1076.8 | -0.25 | 1.45 | -1.70 |
| Clearing event | 2 | 4 | 59.4 | 0.23[*] | 1.95 | -1.73 | 59.8 | 0.38[*] | 1.80 | -1.50 |
| Wet-dry debris curing, decomposition | 3 | 59 | 143.2 | 0.98[**] | 1.39 | -0.41 | 412.0 | 0.32[**] | 1.53 | -1.22 |
| Dry season pre-burn | 4 | 94 | 0 | 0.34[**] | 0.57 | -0.23 | 2.4 | 0.15[**] | 0.94 | -0.79 |
| Fire emissions late dry | 5 | 22 | 0 | 0.90[**] | 0.76 | 0.0 | 0.0 | -0.01[**] | 0.71 | -0.72 |
| Dry season post-burn | 6 | 67 | 2.2 | 0.31[**] | 0.37 | -0.06 | 64.4 | -0.28 | 0.64 | -0.91 |
| Early wet regrowth | 7 | 80 | 361.0 | 0.03[**] | 0.99 | -0.96 | 345.8 | -0.32 | 1.80 | -2.12 |
| Wet season site prep | 8 | 91 | 701.7 | 0.62[**] | 0.99 | -0.37 | 914.4 | -0.20[**] | 1.67 | -1.88 |
| Dry season final bed prep. and cultivation | 9 | 90 | 0 | 0.29[**] | 0.32 | -0.02 | 10.8 | 0.06[**] | 0.91 | -0.85 |
|  |  |  | (Mg C ha$^{-1}$) |  |  |  | (Mg C ha$^{-1}$) |  |  |  |
| Total post-clearing |  | 507 | 1267.5 | 7.2[**] | 12.8 | -5.6 | 1809.6 | -0.78[**] | 20.7 | -21.5 |
| Total all phases |  | 668 | 2004.1 | 6.0[**] | 21.2 | -15.2 | 2886.4 | -2.1[**] | 28.5 | -30.6 |

[*]Denotes significantly different mean NEE at the 5% level, [**]significant at 1%.
Table 4

| Fuel type | Fuel load (Mg C ha$^{-1}$) | BEF | Carbon content | N:C ratio | EF CO$_2$ | EF CH$_4$ | EF N$_2$O | Emissions (Mg CO$_2$-e ha$^{-1}$) | | | |
|---|---|---|---|---|---|---|---|---|---|---|---|
| | | | | | | | | CO$_2$ | CH$_4$ | N$_2$O | Total |
| Fine | 1.1 ± 0.70 | 0.95 | 0.46 | 0.0096 | 0.97 | 0.0031 | 0.0075 | 3.9 | 0.1 | 0.04 | 4.0 |
| Coarse | 0.5 ± 1.0 | 0.9 | 0.46 | 0.0081 | 0.92 | 0.0031 | 0.0075 | 1.5 | 0.0 | 0.01 | 1.6 |
| Heavy - AGB | 26.2 ± 7.0 | 0.9 | 0.46 | 0.0081 | 0.87 | 0.01 | 0.0036 | 75.2 | 7.9 | 0.32 | 83.4 |
| Heavy - CWD | 1.4 ± 0.6 | 0.9 | 0.46 | 0.0081 | 0.87 | 0.01 | 0.0036 | 4.0 | 2.7 | 0.11 | 28.5 |
| Heavy - BGB | 9.0 ± 2.4 | 0.9 | 0.46 | 0.0081 | 0.87 | 0.01 | 0.0036 | 25.7 | 0.0 | 0.02 | 4.4 |
| *Total* | | | | | | | | *110.2* | *11.1* | *0.50* | *121.9* |

Table 5

| Savanna region | Savanna burning | | | Savanna deforestation | | | |
| --- | --- | --- | --- | --- | --- | --- | --- |
| | Burnt area[a] | Emissions non-$CO_2$[a] | Emissions total[a] | Deforestation area | Emissions total | Expanded deforestation area[d] | Expanded emissions total[d] |
| | (ha y$^{-1}$) | (Gg $CO_2$-e y$^{-1}$) | (Gg $CO_2$-e y$^{-1}$) | (ha y$^{-1}$) | (Gg $CO_2$-e y$^{-1}$) | (ha) | (Gg $CO_2$-e y$^{-1}$) |
| Douglas-Daly River catchment | 2,482,100 ±490,400 | 577 ±124 | 14,270 ±3064 | 1275 ±454[b] | 163 ±162[b] | **20,000** | **756** |
| Northern Territory | 13,419,410 ±487,300 | 3,490 ±922 | 86,255 ±22,880 | 1,717 ±611[c] | 398 ±128[c] | **114,500** | **3,413** |
| North Australian | 32,249,254 ±11,176,004 | 6,740 ±1,729 | 166,586 ±42,725 | 78,605 ±34,976[c] | 16,161 ±5601[c] | **311,000** | **24,393** |

[a]Burnt area and emissions data estimated using the on-line Savanna Burning Abatement Tool (SAVBat2), 1990-2013. These emissions are $CH_4$ and $N_2O$ only.
[b]Deforestation area data taken from Lawes et al. (2015), upscaled using the emissions from the CS site from this study, 1990-2013
[c]Deforestation area and emissions data taken from the State and Territory Greenhouse Gas Inventories (Commonwealth of Australia, 2015a), 1990-2013
[d]Expanded deforestation area data taken from catchments as identified by Petheram et al. (2014), upscaled using the GHG emissions from the CS site from this study and added
to historic emissions
**Figure captions**
Figure 1 Location of the uncleared site (UC) and the cleared savanna (CS) sites south of Darwin,
Northern Territory. The inset figure shows the distribution of the savanna biome across northern
Australia as defined by Fox et al. (2001).
Figure 2 Comparative meteorology and fluxes for the uncleared (UC) and cleared savanna CS sites
prior to the clearing event. Data spans the late dry season (September 2011) through to the mid-wet
season prior to the clearing event of 2-6 March 2012. Plots include a) daily precipitation (black bars
UC site, grey bars CS site), mean daily $T_{air}$ (black lines UC, grey CS), b) mean daily VPD (dashed
lines; black UC, grey CS), c) interpolated 8-day MODIS LAI (black UC, grey CS), d) NEE (black
UC, grey CS) partitioned into $R_e$ (red UC, pink CS) and GPP (dark green UC, pale green CS).
Figure 3a) Daily precipitation and b) diurnal patterns of NEE at the CS site for the week prior to the
clearing event of 2-6 March 2012 (vertical bar) and three weeks post-clearing.
Figure 4 Cumulative NEE from the CS (red line) and UC sites (black line) for each land use phase
(see Table 2 for details) over the entire observational period, September 2011 to July 2013. The UC
site is a long-term savanna site of the Australian flux network (OzFlux, see Beringer et al. 2016a)
and using the sites' 8-year flux record (2007-2013), the long-term cumulative mean NEE is plotted
for each land use phase of (grey line; ± 95% CI). The dashed line indicates zero net $CO_2$ flux.

 **Plate caption**


Plate 1 Key LUC phases associated with: a) the clearing event, Phase 3; b) debris burning of the
cured grass, litter and woody fuels following the 5 month curing period, Phase 5; c) stockpiling and
ignition of remaining unburnt debris and d) post-fire site preparation with all biomass consumed,
Phase 9.



Figure 1

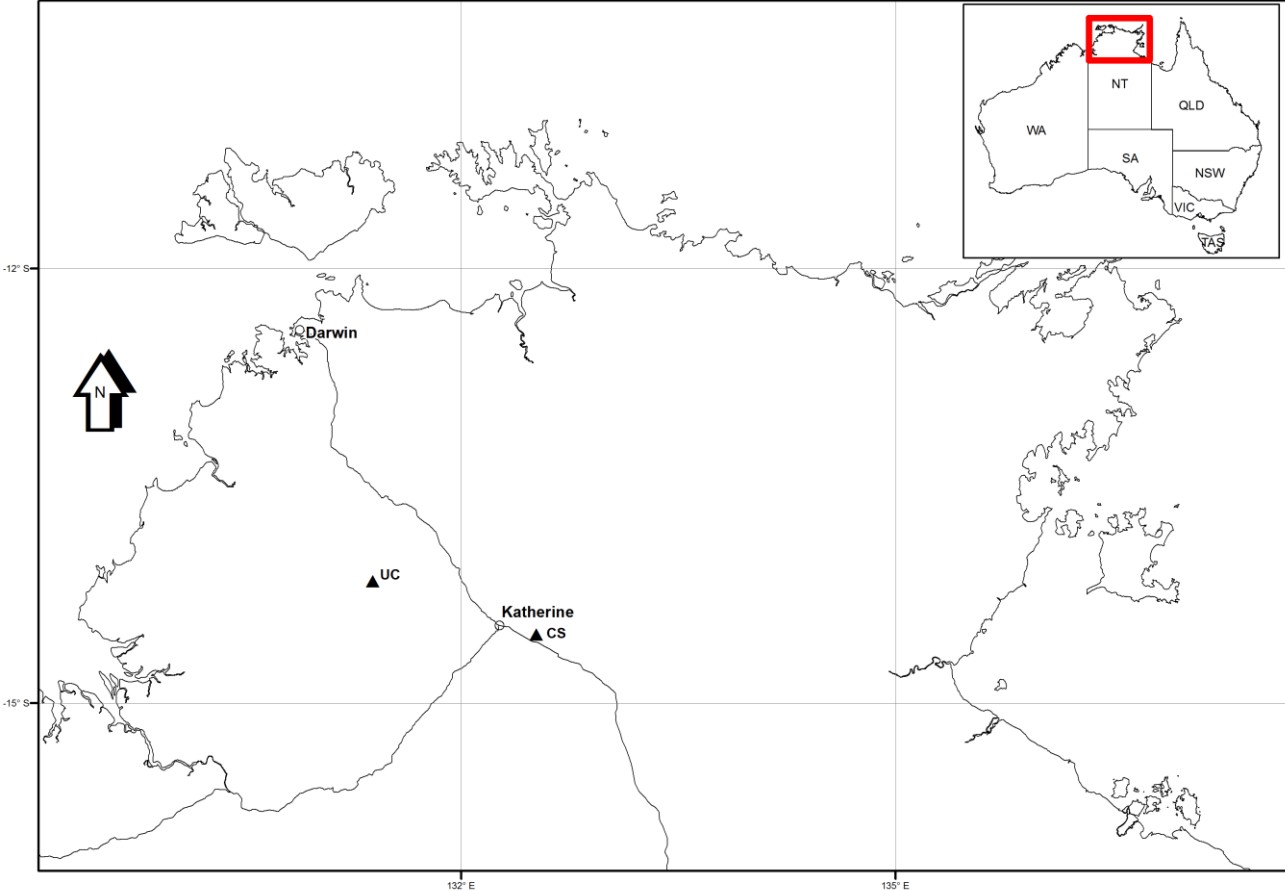



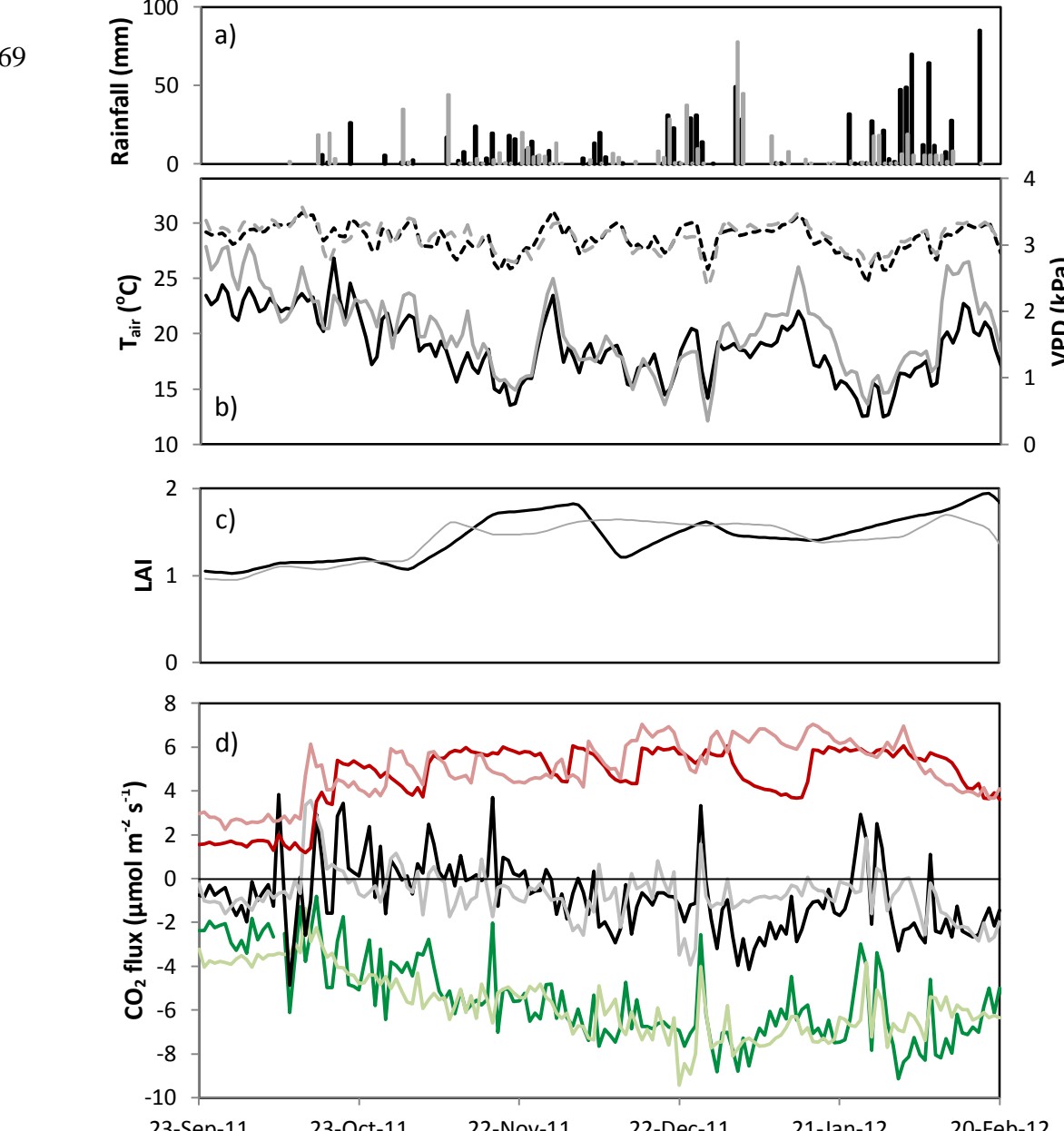

Figure 3



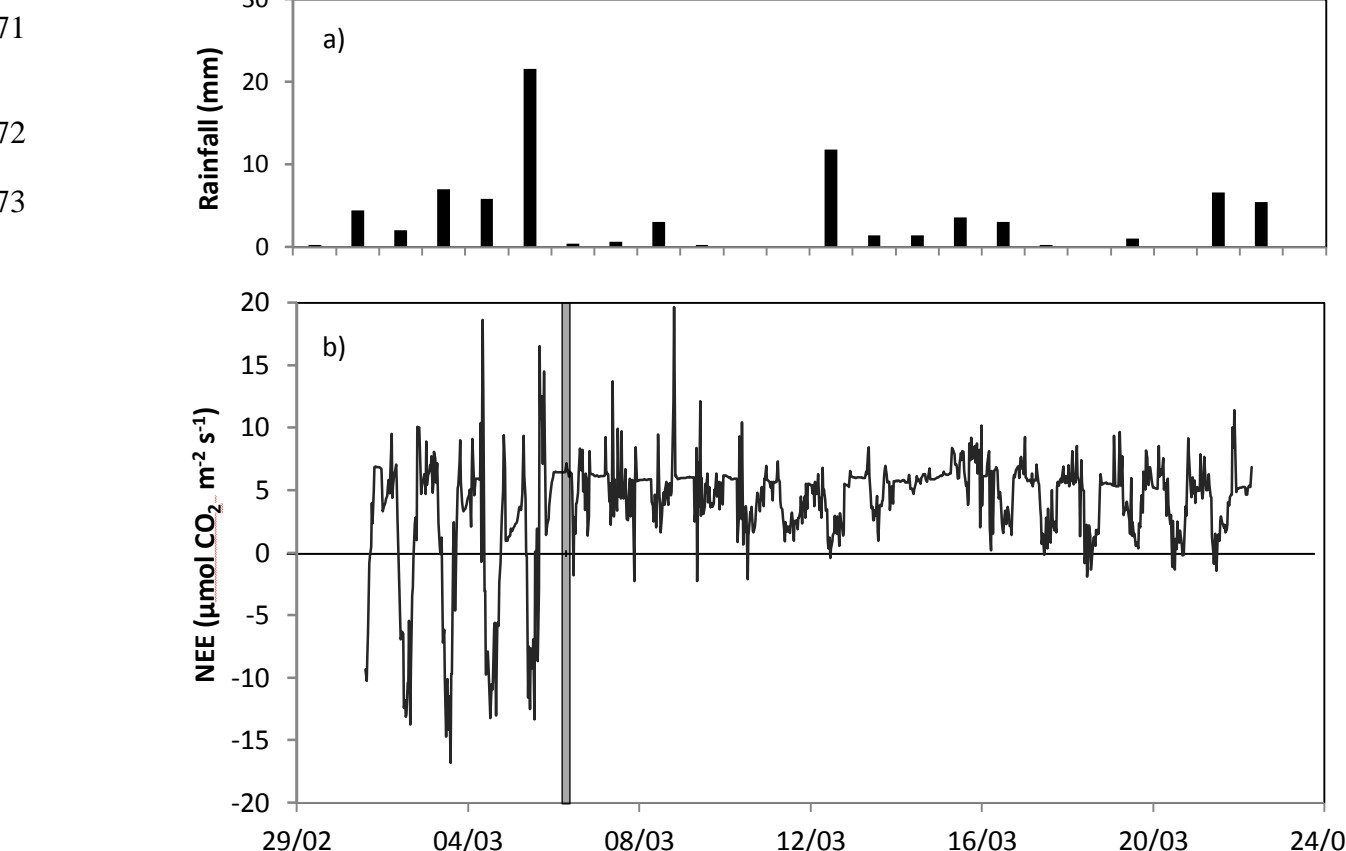

Figure 4

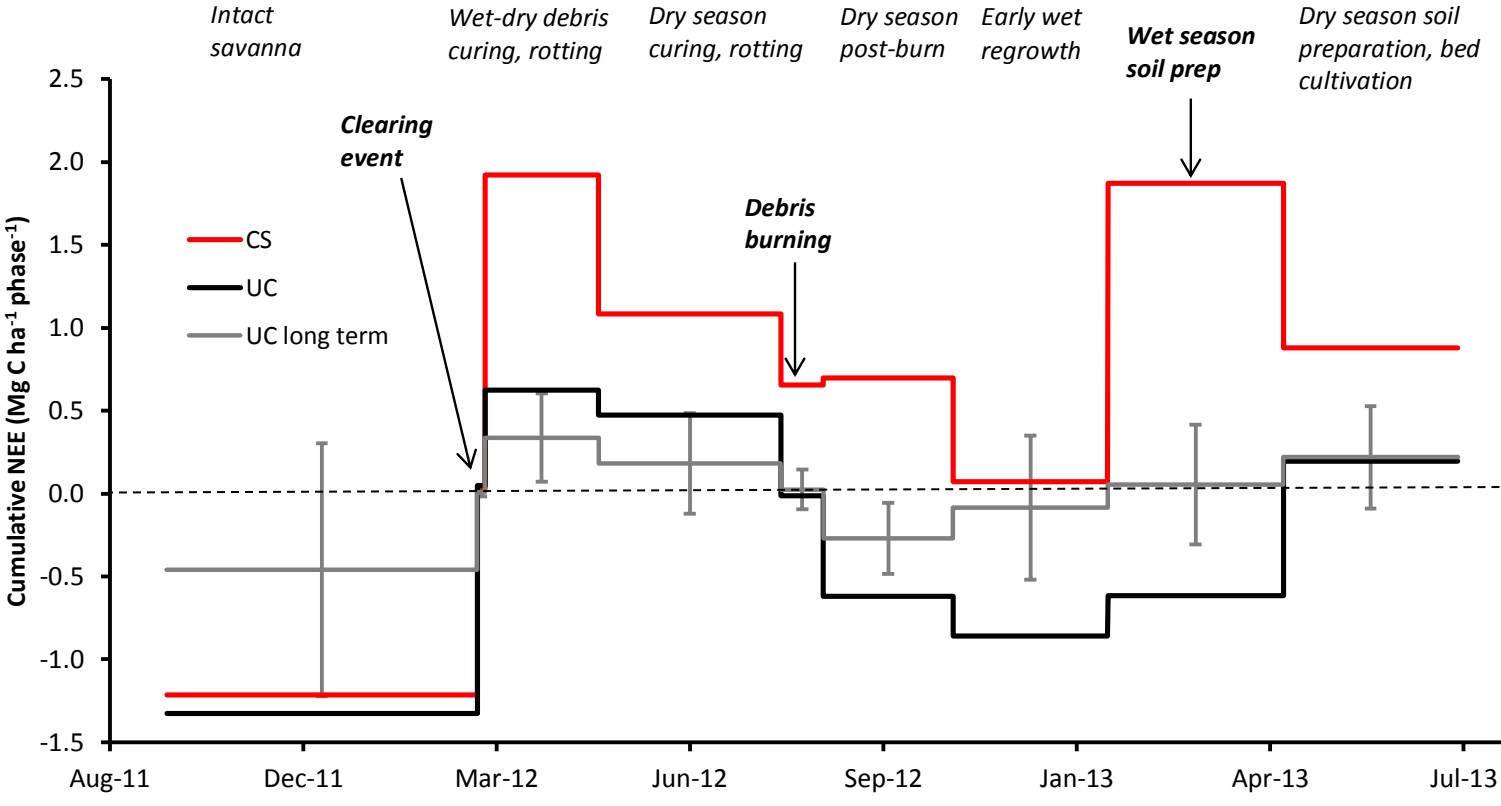


Plate 1

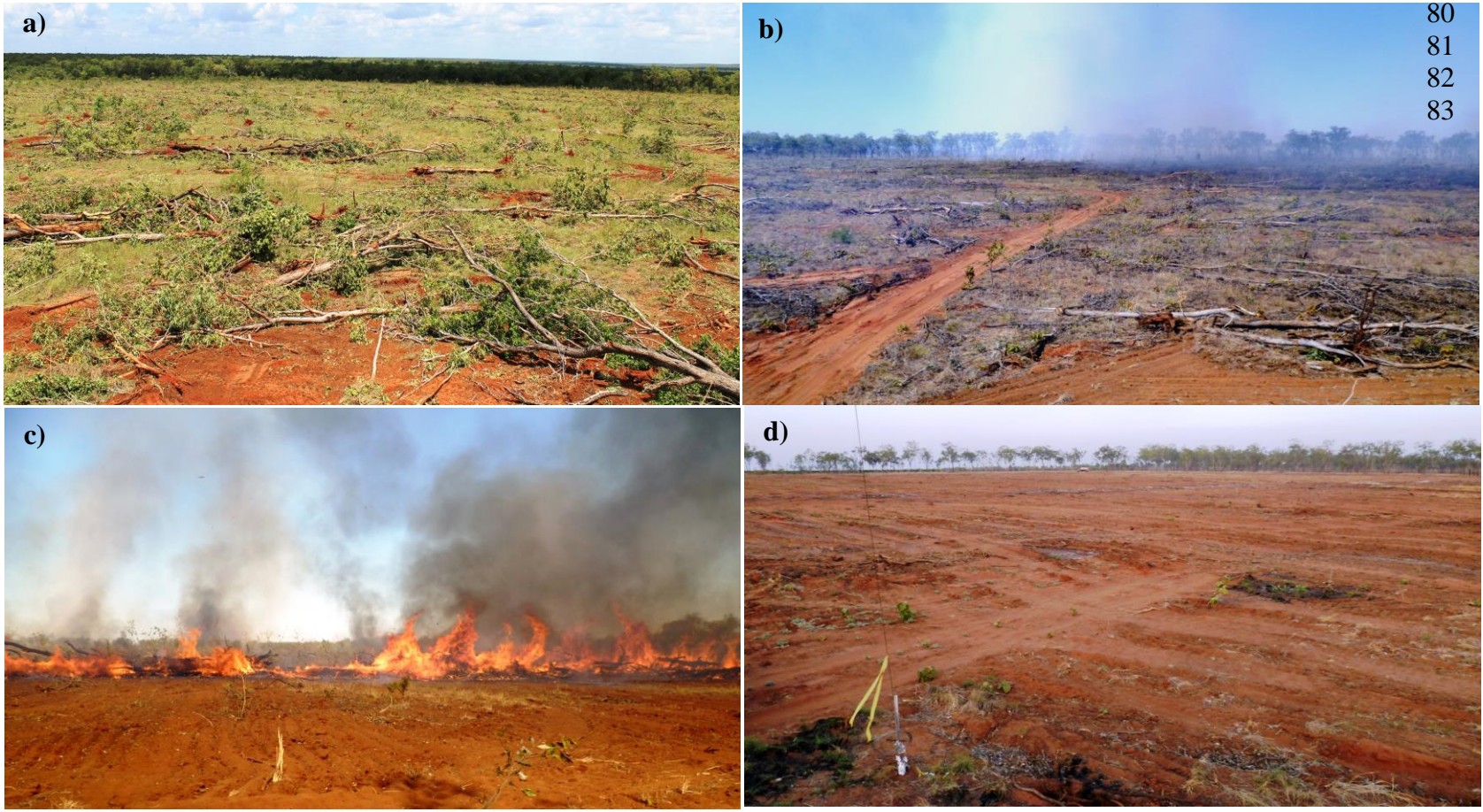