# Peer review of "Quantifying the relative importance of greenhouse gas emissions from current and future savanna land use change across northern Australia"

_Biogeosciences, 2016_

## Referee Comment (RC1) · Anonymous Referee #1 · 28 May 2016

This paper presents very interesting data from flux towers comparing an intact savanna with an adjacent site during a clearing event. Unfortunately the authors do not emphasise the strengths of their work, which is the detailed time series that can compare processes for the two sites. Rather they attempt to extrapolate the results across northern Australia in ways that are not transparent and appear to have a number of flaws.

Page 2 L4: elsewhere you say 30% of Australia. Also (a bit pedantic) the Savannas aren't going to influence national GHG budget, human land use decisions are. L 13: debris were... L 16: see comments in relevant section Page 7, L 5: "...we measured fluxes of....on a single paired site comprising an uncleared savanna and a ...."

Extrapolating such a study is problematic from a statistical point of view. Please comment. L13: Firstly, the original reference (Hurst et al 96) should be cited, secondly that reference is inappropriate for the application you have put it to. It was a study of a natural savanna, where the fuel was predominantly grass and leaf litter, not woody debris which dominates your cleared site. The values for wood in Meyer et al. 2015 (in Carbon accounting etc by Murphy et al.) would be better, but even here they were not measured under extreme conditions of a raked up cleared site. Page 8 L 7 on. You are restating most of the information in Table 1. This is unnecessary. Further the vegetation description differs from that in the table. Can you include estimated biomass of each site broken into above ground etc? Page 9, L 3: In Table 1, you say that fire frequency is 0.3, but 5 in 13 years is 0.38. Why the difference? Background for CS would be the similar rather than 0? Even during the measurement period the CS was burnt twice albeit both in the same year! L 8: you don't mention first stockpiling in Table 2. Page 12, L 13, again these emission factors are inappropriate for a stockpiled log fire. L 18: you cites two allometric papers. Which did you use? It sounds like Williams et al. Page 13, L8: what did you do with the estimates of missing biomass? It would be inappropriate to take it away from allometric estimates because these were surely done on trees with hollows? It would be inappropriate to add it back in because it could have gone decades ago. L20: A Byram fire intensity value has meaning for combustion of fine fuel, not for logs. L 20/21: Meyer et al 2012 (JGR 117) showed that emission factors were not affected by season of burn within the dry season, so the logic of your argument is invalid. Rather, the bigger difference is between fuel types (Meyer et al 2015). This make your choice of Hurst et al values used by Russell-Smith et al 09 invalid. P14 L2: blew away where? Into the next plot? How far? What proportion? L 14-17: where were these data from to estimate these values? C of A 2015? Page 15 L1-5: It is not clear how you intersected single values for each state/territory with the savanna area. Next paragraph: again, I cannot see how these numbers were derived. It is not at all clear where 78605 ha /yr came from. Page 18, L22: what are the four pools? How is the CWD pool (line 22) different to the AGB of line 21? Page 19 L2:

didn't you measure this? Can you estimate it from your flux data? The burning efficiency values of RS et al 09 are not appropriate to use for a fire that was in raked fuel and burnt for 10 days!!!. Line 21: Most clearing was in the 1980s during the ADMA scheme which failed. In recent years there was a clearing moratorium. It is no wonder values from 1990s are low. Line 23: Including $CO_2$ is not consistent with Australian GHG inventory or with your own conclusions in line 7/8 Page 24. Page 20: L1: I still don't see how this figure was calculated and what components it includes. Line 2: "savanna burning emissions" P22, L 14 on: decades of accumulation? L 21: four decades of accumulation? Are you seriously suggesting a stand replacement event happened c. 1973? Cyclone Tracy did not go that far inland. It is hard to imagine floods killing a stand of E. Tetrondonta. Stand killing fire in E tet vegetation are extremely rare unless invaded by African grasses. Was it cleared then? Do you have any evidence? Or is it just a case of slow in fast out cycles and you have just done your measurements during a slow in phases and incorrectly extrapolating it to decadal time scales? Page 24 L5: One page 9, L 5 you say return interval is 3.1 yr or frequency of 0.32. The range you give here is not consistent with that mean value. Line 14/15: I don't see the logic here. In the burning of savannas that remain savannas (ie not cleared), then it is reasonable to assume that the $CO_2$ is taken up by next year's growth. However for a land-clearing event burning with clearing removes substantial carbon from the land and puts it into the atmosphere. This is a loss that needs to be accounted. Not sure what you are trying to say here. Page 25 L3: I just checked the 2013 data on total reportable emissions and LUC emissions and can't see where you got this figure from. It should be much higher. If you read Cook et al. 2010 Rangeland Ecol and Management, you will see that land clearing has been up to 30% of total emissions. Also when reporting % changes it can get confusing with % of what? Might be better using Gg units. References: there are many references (well some), that are cited but not in the ref list. Please check thoroughly.

Table 1: What is the meaning of Blain in soil type (I know, but few others would without a reference). The fire frequency for the CS would be the same as for the UC for the

past decades surely? Even during the observation period it had two fires not zero. Also 5 fires in 13 years is 0.38 not 0.3. Can you add above ground (or below ground) biomass or carbon stock? Table 4: For the deforestation area, it appears you have added the expanded Pertheram et al. values to the background values. Firstly is this valid given background clearing may be in same areas identified by P et al. Secondly, I still don't get how you intersected the state inventories with your savanna area. Thirdly, what is the Standard deviation. Is it year to year variation? What is the validity of this value if you have added a constant (P et al /5) to the rate? The emissions from deforestation seem low. In estimating emissions, how were the biomasses across the region estimated? Figure 2: The fonts are far too small to read

---

## Referee Comment (RC2) · B. Amiro (Referee) · 27 Jun 2016

Review of Bristow et al. Biogeosciences OzFlux 191

General Comments:

This paper describes flux tower measurements of carbon dioxide for a paired treatment where one site was disturbed while another was used as a control. The paper provides new data on comparative net ecosystem exchange. It further estimates GHG emissions from a burning activity during land clearing, and provides a perspective on broader implications over a large area of northern Australia.

The science is basically sound and the experiment is a worthwhile contribution. The

manuscript needs to be carefully edited before acceptance. Some specific comments are given below, many of which are editorial. However, there are several sections that require additional technical input.

Specific Comments:

1. Page 2, line 10. No need for the "EC" abbreviation in the abstract. 2. Page 3. Line 9. Remove "northern" since it is not needed and misleading for a global audience. 3. Page 4, lines 3 and 5. The references for IPCC and Tubiello are not in the reference list. 4. Page 4, lines 22-25. These lines are repeated. 5. Page 5, Line 8. "Leakage" is not clear; suggest to avoid the term. 6. Page 6, lines 3-4. "Significant" used twice. Also put "in" preceding "northern". 7. Page 6, line 8. Extra bracket in reference. 8. Page 8, line 15 and elsewhere. "SD"instead of "sd". 9. Page 8, lines 20 and 23. The heights and towers are not clear. Perhaps say height-adjusted because only the instrument height is important, not the tower height. 10. Page 9, line 5. Remove "one in" to make the return time correct. 11. Page 9, line 13. Why say GHG when this part is all about carbon? 12. Page 9, line 22. Give address for first mention of a company. 13. Page 10, line 1 and elsewhere. The Ozflux convention uses subscripts for the energy balance components. Also Rn is used for net radiation here but then Fn is used on page 11, line 4. 14. Page 10, lines 4 to 16. The method for flux calculation is not clear. The Isaac paper says that the high-frequency data are recorded and then processed. But this section says that only the 30-min covariances were recorded, so some of the procedures, such as spike removal, could not be applied. Please rewrite this section so that the audience can follow the method. 15. Page 11, line 8. This is repeated in Lines 17-20 on Page 10. 16. Page 11, lines 14-16. The gap-filling method is not clear. In the Results, the different phases are described. Because the fluxes can be very different among phases, it would seem that the gap-filling neural network should only be trained for each phase. It is not clear that this was done. Further, the amount of gaps are not presented, and the use of different ANN for each phase needs to be evaluated to identify any artefacts caused by the gap filling. 17. Page 13, line 1. It would appear

that emission factors for burning were developed for wildland fires, and not for piled, cured fuels. Please give us an idea of the possible error in the emission factors. I realize that you have suggested that the 10% residual following burning is similar, but this would only tell us about carbon, not about the non-CO2 emissions. 18. Page 14, lines 3-6. These two sentences seem to say the same thing. 19. Page 14, lines 3-6, and Table 3 (fire emissions late dry). It appears that the flux tower data were not used while the area was burnt to exclude combustion emissions in the NEE measurement. Please confirm. If so, where did the value (0.90) for the CS tower come from? Was it the NEE measurement without a smoke plume? 20. Page 15, lines 3 and 10: check grammar. 21. Page 16, line 12. Figure 2 does not add much to the paper because there really is no energy balance component to the study, just GHG. So remove Figure 2, and just describe the general nature of energy balance closure in the text. 22. Page 17, line 10. I think you mean Fig 4 here, not 3, for this period. 23. Page 18, line 6. "Mg" 24. Page 18, line 16. Figure 5 shows an amount less than 2.75; where did this come from? 25. Page 19, lines 17-18. Awkward sentence. 26. Page 20, lines 6 and 8. The values given in the text are not the same as those in Table 4. 27. Page 20, line 23. Other GHG gases were not measured; omit the part in brackets. 28. Page 21, line 3. Spelling. 29. Page 22, line 15 and elsewhere in manuscript. Proof-read carefully everywhere. For example insert "be" ahead of "a weak". 30. Page 23, line 6. Any evidence for this statement? In temperate systems, cropping systems are usually a carbon loss once harvest is included. 31. Page 24, line 15. "24,769" is not in Table 4. 32. Page 24, line 9. The greatest error is often in gap-filling, and you did not assess this part in the study. We need further information to assess this. 33. Page 25, lines 16-19. Very awkward. 34. Page 26, line 5. "ranging". 35. References: check very carefully. For example: Page 29, line 32. Should be Paw U, K.T. 36. Table 3. NEE is given in Mg per month, but the totals are total Mg. Perhaps insert a new row of units for the two "total rows" so that the reader can see the unit change. Also, it would have been good to have done a statistical comparison between CS and UC for each phase and show here if different (days would be replicates). You did this only for the intact

period. This would make the paper more experimental to test a hypothesis. 37. Figure 3 caption for d). Say this is daily average NEE. 38. Figure 1. Latitude and longitude lines would help the audience. 39. Figure 3 axis for VPD should be kPa instead of KPa.

---

## Editor Comment (EC1) · B. Amiro (Editor) · 27 Jun 2016

Special Issue Co-Editor comments on Bristow et al. "Quantifying the relative importance of greenhouse gas emissions. . ." Biogeosciences Special Issue on OzFlux paper 191.

Special Issue Co-Editor: Brian Amiro, University of Manitoba, Winnipeg, Canada.

General Comments and Recommendation

This manuscript received input from one anonymous reviewer. As special issue co-editor, I have provided the second technical review.

[Figure]

Reviewer 1 has focused on the applicability of the broader calculations and assessment of the greenhouse gas emissions from land disturbance and fire on the Australian landscape. Many of the issues are related to traceability of the calculations and lack of clarity in data selection. I urge the authors to address these comments by better description (and perhaps modification) of their broader landscape analyses in the revised manuscript. Some of these changes may require recalculations and change the conclusions.

Reviewer 2 (Special Issue Co-Editor Amiro) focused more on the measurement part of the manuscript. Most of these changes should be accommodated in the revised manuscript.

Given the nature of these comments, the manuscript status is accepted with major changes.

---

## Author Response (AR1)

Reply to reviews – Bristow et al bg-2016-191  Greenhouse gas emissions from savanna land use change across northern Australia

We thank both reviewers for their comprehensive assessment of our paper.

Reviewer 1 (R1)

R1 contended that the ms provided very interesting data from the paired flux tower sites but suggested we did "… not emphasise the strengths of their work, which is the detailed time series that can compare processes for the two sites. Rather they attempt to extrapolate the results across northern Australia in ways that are not transparent and appear to have a number of flaws"

This was a reasonable criticism but this ms is part of the OzFlux Special Issue and here we wanted to highlight the use of flux observations in refining GHG emissions and impacts on national accounts, a management focus rather than a detailed physiological analysis.

The flux observations were used to highlight respiration differences between sites, the magnitude of soil $CO_2$ emissions during tillage and preparation for cropping, the continued uptake of carbon post-deforestation from grass growth and woody re-sprouting, as well as the net loss of the natural C sink that we observed at the uncleared analogue site.

Other papers in the Special Issue have documented the flux characteristics of north Australian savannas – data from four savanna flux towers were included in the overview paper of Beringer et al. 2016 (doi:10.5194/bg-2016-152).

R1 provided very useful comments aimed at improved descriptions and estimation of fuels and associated emissions from the debris burning post deforestation, an important component of the total emission from this LUC. As a result the Methods section has been re-written and restructured and broken into 8 sections instead of 6. The revised ms also features a new Table, Table 3, that describes in a more transparent manner, the distribution of fuels as measured across the deforested site using fuel classifications as defined by the Australian Government's savanna abatement methodology and latest emissions factors. Table 3 includes data for each fuel type (fine, coarse, heavy) including fuel mass and the associated emissions factors for each GHG ($CO_2$ $CH_4$ and $N_2O$), N:C and %C content with the emissions calculation described by an equation, Equation 1.

The emission calculation is now based on emission factors as recommended by R1, namely those of Meyer et al. 2012 and the Savanna Abetment Methodology Determination, March 2015, a methodology that is now legislated by the Commonwealth government (www.legislation.gov.au/Details/F2015L00344). This is a more robust and transparent reporting of emissions from the debris fire, although the new estimate (121.9 Mg $CO_2$-e ha$^{-1}$) differs by only 5% (now smaller) when compared to the original estimate. The new estimate has a higher contribution from non-$CO_2$ fluxes.

These improvements are in response to R1's comments arising from text on P7 to P12 of the original ms and we believe this has improved the clarity of the Methods section.

We suspect R1 (and R2) assumed we had one or two large stockpiles of heavy fuels that burnt very hot for 10 days or more. This was not the case as burning of the site consisted of ignition of the cured grasses and woody fuels in situ, with no stock-piling as an initial phase of burning. To ensure safety, ignition was done in blocks, which is why the process took 20+ days. After an intial burn, unburnt woody debris was then stockpiled and re-burnt until incinerated in a second phase. As such we had multiple debris piles distributed across the 295 ha block as opposed to one or two large piles burning at very high temperatures. Again, this has been described more fully in the revised text and to aid in the description of the LUC phases, in particular the fire event and aftermath, a colour Plate, Plate 1, has now been included.

Plate 1 consists of 4 images of the site showing the initial deforestation event, the debris fire and stockpiling and the finalised state of the site prior to bed preparation for cropping.

Descriptions of the data sources for savanna-specific deforestation emissions across north Australia have been improved as R1 (and R2) found this hard to follow. All data were sourced from the NGGI in collaboration with staff from the Commonwealth government's departmental reporting team, as was acknowledged in the Acknowledgements. References to the methodology are given. In short, the savanna boundary that was defined by Fox et al. 2001 was applied as a spatial 'mask' to constrain the area of emissions estimates to the savanna region only. These were then compared to emissions from savanna burning from the same area.

R1 queried how the value of 78, 605 ha $y^{-1}$ in Table 4 (now Table 5) was derived. It is described in text in the Methods section, in Table 4 (5)'s caption and footnote for the table - this value is derived from the savanna constrained deforestation area and is the mean savanna area defrosted per year 1990-2013.

R1 also queried our LUC emission figure for 2013 and suggested our reported value was low. Firstly we report the mean from 1990 to 2013 and secondly we are not reporting emissions from all activities with the LUC sector – only emissions from Activity 2 'Deforestation' are relevant to our study. This was indicated in the original text. Plus our emissions estimate is specifically limited to the savanna land area across WA, NT and Queensland. If R1 looked up reportable emissions for these states in their entirety, there would be a significant difference compared to our reported value as the standard data reported by the Commonwealth includes the non-savanna (non-tropical) areas of each state where there have been significantly higher deforestation rates. Comparing with Cook et al 2010 may be problematic as the area included in each study would need to be identical, especially areas of Queensland, which have experienced significant clearances in southern and central Queensland, outside of our study area. It should also be noted that for the regional savanna estimates, we are simply compiling emissions data for either savanna burning or deforestation as estimated by the Commonwealth, but constrained to the savanna area as defined by Fox et al.

There were a number of other inconsistencies R1 commented on and these have all been addressed: fire frequency data were inconsistent and have been corrected; the citation for biomass allometry is confined to Williams et al. 2005; reference to fire-line intensities has been deleted given our fuel load is a mixture of grass and heavy fuel, with heavy fuel dominating.

We have also improved text in the Methods describing CWD estimation, in particular dealing with hollowing of large CWD fragments – we do not 'add missing biomass' as R1 queried, our method is designed to estimate the missing *volume* to ensure we do not provide a large overestimate of CWD. This was not entirely clear in the original ms and the text is now improved. We are estimating the volume for each CWD fragment that is then converted to biomass using specific wood densities assigned to our 5 rot classes that we define.

R1 also commented on our text re stand replacement events such as cyclones and/or floods which would take 4 decades to recover the lost carbon. The original text was confusing as given the site locations, neither of these events / scenarios is feasible and this sentence has been modified accordingly. The only agent of stand replacement in the region of our sites would be deforestation and conversion for agricultural production.

We include both $CO_2$ (not reportable) and non-$CO_2$ emissions (reportable) for savanna burning for comparison with deforestation emissions.

Reviewer 2 (R2)

Comments by R2 related to improvements in expression and typographical errors throughout the ms as well as an inclusion of a statement of potential errors.

All suggested changes of R2 have been implemented.

Fig 2 on energy balance closure was removed as suggested by R2 and text describing slope statistics from the closure analysis has now been incorporated included in the revised text.

R2 queried the nature of the gap filling approach used – a unique ANN model was developed for each LUC phase given the significant change in canopy and microclimatic characteristics of each phase. Text describing this has been improved. Errors associated with gap filling using the DINGO system were minimal as we had less than 10% of data that was missing.

In this study we used 30 minute covariance data for the calculation of fluxes not the raw data as is inferred from the paper of Isaac et al. in the Special Issue.

We can confirm fire emissions were not included in the NEE measurements and the value of 0.9 (BEF) was derived from an assessment of remaining heavy fuel levels post-fire. This value influences calculations of both $CO_2$ and non-$CO_2$ as described in the new methods section and the new Table 5 that gives emissions factors.

R2 queried the value of 2.75 Mg ha$^{-1}$ – this is the combined emission from the soil tillage phases over the last 6 months of the measurements, as is described in the text.

As requested by R2, a statement on potential errors associated with our emissions estimate from the debris fire has been included, which is based on uncertainty measures as described Russell-Smith et al. (2009) for key parameters used in the Australian savanna emissions methodology. Given a number of key parameters were measured on site in this study, with fuels measured across a well-defined area, our errors will be relatively low when compared with catchment scale to regional scale projects that the methodology has been designed for.

As R2 suggested, we statistically tested for site differences for each LUC phase (1-way ANOVA) with all phases significantly different except the pre-clearing phase, Phase 1. Significantly different mean NEE are identified in Table 3.

Figure 1 has been improved as requested, with latitude and longitude lines marked and a higher resolution coastline used.

We thank both reviews for their very constructive comments on the paper.

[revised manuscript text omitted]
 involves pulling trees over using a large chains dragged between two bulldozers, followed by the mechanical stockpiling of woody debris to decay and cure prior to burning. This is followed by raking and stock-piling of any remaining debris and re-burning. Finally, there was mechanised ripping of soil to remove remaining coarse root material to 60 cm depth. These processes result in the removal of all above-ground and most of the below-ground biomass, such that the soil was ready for tillage and cultivation. These phases result in a series of events that may lead to short-term, pulsed GHG emissions that would otherwise be missed or greatly under-estimated by episodic measurements taken at a weekly or monthly frequency after an initial tree felling event (Neill et al., 2006; Weitz et al., 1998). The CS site was cleared between 2 and 6 March 2012, which is towards the end of the wet season; 737 mm of rainfall had fallen since the end of the preceding dry season. Over this five day period, 295 ha of savanna were cleared. 
[revised manuscript text omitted]
 north Australian savanna (Murphy et al., 2015a) for a range of fuel types (grasses, fine and coarse woody fuels, Russell-Smith et al. 2009). Fuel load (biomass) estimates are essential to this approach and was quantified for four fuel types prior to clearing: 1) above-ground woody biomass, 2) below-ground biomass, 3) surface coarse woody debris (CWD), and 4) grass biomass. Emissions estimates were based on fuel mass per area for each fuel type, carbon content (%), elemental C:N ratios and Australian savanna combustion emissions factors for $CH_4$ and $N_2O$ (0.0035, 0.0076 respectively, Russell-Smith et al., 2009b) and $CO_2$ (0.87, Hurst et al., 1994).

[revised manuscript text omitted]

 savannas which was 0.38. ~~These savanna trees have no dominant tap root, but large lateral roots in the top 30 cm of soil and up to 90% of root biomass occurs in the top 50 cm (Eamus et al. 2002). As such, we assumed that chaining and bulldozer clearing of all above-ground biomass and soil ripping (ploughing) to 60 cm soil depth, plus mechanised removal of root biomass associated with tree bole, resulted in a near-complete removal of both above- and below-ground woody biomass pools. This debris was subsequently stockpiled for curing over the dry season and then burnt (Table 2).~~

~~The CWD pool was defined as any fragment with a diameter >6 mm and was estimated using a line intercept method (Woldendorp et al., 2004). Six 100 m transects, randomly located across the cleared block, were established and along each transect the length and diameter of all intersected CWD fragments were recorded. Large fragments (>100 mm diameter) are frequently hollowed from the action of termites and fire, and the diameter of the annulus and fragment were measured to estimate this missing biomass.~~

 were defined as recently fallen, solid wood (RC1), solid wood with or without branches present but with signs of aging (RC2), obvious signs of weathering, still solid wood, bark may or may not be present (RC3), signs of decay with the wood sloughed and friable (RC4) and severely decayed fragments with little structural integrity remaining (RC5). A wood density was assigned to each RC and species (where identifiable) after Rose (2006) and Brown (1997) to provide an accurate estimate of CWD mass that included decay and hollowing. For the dominant *Eucalyptus* and *Corymbia* species wood densities ranged from 0.7 g cm$^{-3}$ (RC1) to 0.56 g cm$^{-3}$ (RC 5).

Above-ground biomass was quantified by surveying all woody plants >1.5 m in height or > 2 cm DBH across eight 50 x 50 m plots. All woody individuals were identified to species and stem diameter at 1.3 m height (DBH) and tree height were measured. Region specific allometric equations are available for tree species found at the CS site (Williams et al., 2005) and these were used to estimate above-ground biomass for each individual tree and shrub based on DBH and height. Below-ground biomass was calculated using the root:shoot ratio estimate of Eamus et al. (2002) for these ~~Debris from the March 2012 clearing event was allowed to cure for months through the dry season to ensure a high fire intensity (>5 MW m$^{-1}$) and combustion efficiency. This period is of similar duration that fuels can naturally cure in this landscape, enabling the application of the regional savanna emission factors as defined by Russell-Smith (2009b). Burning of stockpiled debris lasted approximately 10 days in August 2012, the late dry season (Table 2), a period of high winds and low daytime relative humidity (10-20%). Debris that did not burn was stock-piled for a second time and burnt to ensure all biomass was consumed with ~10% remaining as ash and charcoal, a typical fraction from high severity fires (~~savanna stands which was 0.38. These trees have large lateral roots in the top 30 cm of soil, with no tap root and 90% of root biomass is found in the top 50 cm of soil (Eamus et al. 2002). As such, we assumed that chaining and bulldozer clearing of all above-ground biomass followed by soil ripping (ploughing) to 60 cm soil depth, plus mechanised removal of root biomass associated with tree boles and subsequent burning, resulted in a near-complete removal of both above- and below-ground woody biomass pools (Plate 1d).

**2.7 Deforestation**

fluxes as captured by the flux tower plus $CO_2$ and non-$CO_2$ ($CH_4$, $N_2O$) emissions from debris burning.

**2.6 Emissions from deforestation and savanna burning emissions at catchment to regional scales**

The potential impact of any expanded deforestation across north Australian savanna landscapes was assessed relative to historic deforestation rates and resultant GHG emissions and arising from prescribed savanna burning. This land management activity  contributes ~3% to Australia's national GHG emissions (Whitehead et al., 2014) and is 25% of the Northern Territory's annual emissions (Commonwealth of Australia, 2015a). Annual emissions from these activities (historic and future savanna deforestation and prescribed burning) were estimated at three spatial scales; 1) catchment, 2) state/territory and 3) regional. Emissions estimates from deforestation and savanna burning were compiled for 1) the Douglas-Daly River catchment where the UC and CS sites are located (area 57,-571 km$^2$), a catchment with less than 5% of the native vegetation deforested to date (Lawes et al. 2015) but earmarked for future development; 2) the savanna area of Northern Territory (856,000 km$^2$) and 3) the savanna region of north Australia  as defined by Fox et al. (2001) with MAP > 600 mm, an area of 1.93 million km$^2$ (Fig. 1, insert).

~~Historic deforestation emissions from the Douglas-Daly catchment were estimated using satellite-derived clearing areas (1990-2013) for the catchment as reported by Lawes et al. (2015). These annual deforestation areas were combined with our estimate of GHG emissions from the CS site to give a mean annual estimate in Gg CO$_2$-e y$^{-1}$. To estimate GHG emissions from savanna deforestation at state/territory and regional scales, data for the Northern Territory and the north Australian savanna region were taken from the reported emissions under Activity A.2 under the Land Use, Land-Use Change and Forestry sector of State and Territory Greenhouse Gas Inventories~~

(Commonwealth of Australia, 2015). Annual reporting of these GHG emissions is state based, not biome based, and for the regional savanna estimate, data for Western Australia, the Northern Territory and Queensland were used but included only the area within each state that was as defined as savanna after Fox et al. (2001, Fig. 1) and emissions from each state were summed to give the north Australian savanna regional estimate.

To estimate the GHG emissions from future expanded deforestation across north Australia, we upscaled our estimate of deforestation emissions per hectare using the areas identified as having future clearing potential following the land use assessment of north Australian catchments by Petheram et al., (2014). This preliminary assessment identified catchments to be cleared based upon surface water storage potential and proximity of land resources suitable clearing for irrigation development to enable high-value farming such as irrigated agriculture, horticulture or improved pasture. Using these criteria, suitable catchments in Western Australia (Fitzroy River, Ord Stage 3; 75 000 ha potential area), the Northern Territory (Victoria, Roper Rivers, Ord Stage 3, Darwin-Wildman River area; 114, 500 ha) and Queensland (Archer, Wenlock, Normanby, Mitchel Rivers; 120 000 ha) were selected. This gives a projected savanna clearing area of 311, 000 ha, equivalent to an additional 16% of cleared land over and above the 1,886,512 ha that has been cleared across the savanna biome since 1990 (Commonwealth of Australia, 2015). Projected emissions calculations included emissions from historic emissions plus additional emissions estimates from the expanded deforestation areas. Emissions from the expanded deforestation areas were calculated assuming any such clearing would occur over a five year period. This filter provided identical areas for comparison of mean annual savanna deforestation and prescribed burning emissions.

[revised manuscript text omitted]
⁻¹ (0.154 Gg CO₂-e ha⁻¹ y⁻¹), with the remainder attributed to the net ecosystem exchange of CO₂ as measured by the flux tower. This comprised significant CO₂ losses via respiration of debris, enhanced soil CO₂ efflux from soil disturbance and tillage, which was partially offset by net uptake of CO₂ from woody re-growth, re-sprouting, grass germination and growth~~

At the CS site, burning of post-clearing debris of comprised 82% of the total emission of 148.4 Mg $CO_2$-e ha$^{-1}$, with the remainder attributed to NEE as measured by the flux tower. This flux comprised significant $CO_2$ losses via respiration of debris, enhanced soil $CO_2$ efflux from soil disturbance and tillage, which was partially offset by net uptake of $CO_2$ from woody re-sprouting post-clearing and periods of grass growth following wet season rainfall (Fig. 4). Soil disturbance via ripping, tillage and preparation was responsible for 10% of the $CO_2$ emission from the conversion. The EC flux tower was operational during the clearing event, demonstrating the utility of this method as the switch of the ecosystem from being a net $CO_2$ sink to being a net source occurred over a number of hours as the clearing event was completed (Fig. 3). During the LUC phase changes, there was little evidence of major pulses of $CO_2$ flux, instead there was a rapid transition to a new diurnal pattern following the clearing (Fig. 3) or the commencement of soil preparation (data not shown). This is in contrast to non-$CO_2$ flux emissions, in particular $N_2O$, with short term emissions often follow disturbance (Grover et al., 2012; Zona et al., 2013) and can be a significant fraction of annual emissions.

The net $CO_2$ source as measured by the flux tower represents an emission that would be missed if vegetation biomass density alone was used to estimate LUC emissions, the approached used in current remote sensing LUC studies at regional and continental scales, data that is the basis of  emissions reporting for the LULUC sector. The total GHG emission we report in this study is more accurately described as a land conversion, as it includes the oxidation of biomass plus emissions associated with soil disturbance and tillage required for a conversion to a cropping or grazing system.

The emission estimate from this study does not include non-$CO_2$ soil derived fluxes of $CH_4$ and $N_2O$, which can be significant for LUC events in certain ecosystems (Tian et al., 2015). Grover et al. ~~(2012) compared soil $CO_2$ and non-$CO_2$ fluxes from native savanna with young pasture and old pastures (5-7 and 25-30 years old) in the Douglas-Daly River catchment. Soil emissions of $CO_2$-e were 30% greater on the pasture sites as compared with native savanna sites, with this change being dominated by increases in $CO_2$ emission and soil $CH_4$ exchange shifting from a small net sink to a small net source at the pasture sites. Non-$CO_2$ soil fluxes were generally small, especially $N_2O$ emissions, although these measurements were made many years after the LUC event and there is uncertainty as to their relevance for a recently deforested and converted savanna site. An additional pathway for $CH_4$ and $N_2O$ emissions in these savannas is through termite activity (Jamali et al., 2011a, 2011b), and in our study, termite mounds were abundant across the CS site, but were largely destroyed by clearing and soil preparation, potentially reducing net non-$CO_2$ emissions. Further work is required to quantify these fluxes and refine our total emission estimate for this LUC event.~~(2012) 
[revised manuscript text omitted]

R., Viovy, N., Wang, Y.-P., Wanninkhof, R., Wiltshire, A. and Zeng, N.: Global carbon budget 2014, Earth Syst. Sci. Data, 7(1), 47–85, doi:10.5194/essd-7-47-2015, 2015.

Macfarlane, C., Hoffman, M., Eamus, D., Kerp, N., Higginson, S., McMurtrie, R. and Adams, M.: Estimation of leaf area index in eucalypt forest using digital photography, Agric. For. Meteorol., 143(3–4), 176–188, doi:10.1016/j.agrformet.2006.10.013, 2007.

Mayocchi, C. L. and Bristow, K. L.: Soil surface heat flux: some general questions and comments on measurements, Agric. For. Meteorol., 75, 43–50, 1995.

Meyer, C. P., Cook, G. D., Reisen, F., Smith, T. E. L., Tattaris, M., Russell-Smith, J., Maier, S. W., Yates, C. P. and Wooster, M. J.: Direct measurements of the seasonality of emission factors from savanna fires in northern Australia, J. Geophys. Res. Atmos., 117(D20), n/a-n/a, doi:10.1029/2012JD017671, 2012.

Murphy, B. P., Lehmann, C. E. R., Russell-Smith, J. and Lawes, M. J.: Fire regimes and woody biomass dynamics in Australian savannas, J. Biogeogr., 41(1), 133–144, doi:10.1111/jbi.12204, 2014.

Murphy, B. P., Edwards, A. C., Meyer, C. P. (Mick) and Russell-Smith, J.: Carbon Accounting and Savanna Fire Management, CSIRO Publishing, Melbourne, 2015a.

Murphy, B. P., Liedloff, A. C. and Cook, G. D.: Does fire limit tree biomass in Australian savannas?,? Int. J. Wildl. Fire, 24(1), 1, doi:10.1071/WF14092, 2015b.

Neill, C., Piccolo, M. C., Cerri, C. C., Steudler, P. a. and Melillo, J. M.: Soil solution nitrogen losses during clearing of lowland Amazon forest for pasture, Plant Soil, 281(1–2), 233–245, doi:10.1007/s11104-005-4435-1, 2006.

Petheram, C., McMahon, T. A. and Peel, M. C.: Flow characteristics of rivers in northern Australia: Implications for development, J. Hydrol., 357(1-2), 93–111, doi:10.1016/j.jhydrol.2008.05.008, 2008.

Petheram, C., McMahon, T. A., Peel, M. C. and Smith, C. J.: A continental scale assessment of Australia's potential for irrigation, Water Resour. Manag., 24(9), 1791–1817, doi:10.1007/s11269-009-9525-z, 2009.

Petheram, C., Gallant, J., Wilson, P., Stone, P., Eades, G., Roger, L., Read, A., Tickell, S., Commander, P., Moon, A., McFarlane, D. and Marvanek, S. .: Northern rivers and dams: a preliminary assessment of surface water storage potential for northern Australia, Canberra, 2014.

Poulter, B., Frank, D., Ciais, P., Myneni, R. B., Andela, N., Bi, J., Broquet, G., Canadell, J. G., Chevallier, F., Liu, Y. Y., Running, S. W., Sitch, S. and van der Werf, G. R.: Contribution of semi-arid ecosystems to interannual variability of the global carbon cycle, Nature, 509(7502), 600–603, doi:10.1038/nature13376, 2014.

[revised manuscript text omitted]
 (d) | Period (d) | CS Rainfall (mm) | CS NEE | CS Re | CS GPP | UC Rainfall (mm) | UC NEE | UC Re | UC GPP |
|---|---|---|---|---|---|---|---|---|---|---|
| | | | | (Mg C ha$^{-1}$ month$^{-1}$) | | | (mm) | (Mg C ha$^{-1}$ month$^{-1}$) | | |
| Intact canopy cover | 1 | 161 | 736.6 | -0.23 | 1.57 | -1.79 | 1076.8 | -0.25 | 1.45 | -1.70 |
| Clearing event | 2 | 4 | 59.4 | 0.23* | 1.95 | -1.73 | 59.8 | 0.38* | 1.80 | -1.50 |
| Wet-dry debris curing, decomposition | 3 | 59 | 143.2 | 0.98*** | 1.39 | -0.41 | 412.0 | 0.32** | 1.53 | -1.22 |
| Dry season pre-burn | 4 | 94 | 0 | 0.34*** | 0.57 | -0.23 | 2.4 | 0.15** | 0.94 | -0.79 |
| Fire emissions late dry | 5 | 22 | 0 | 0.90*** | 0.76 | 0.0 | 0.0 | -0.01** | 0.71 | -0.72 |
| Dry season post-burn | 6 | 67 | 2.2 | 0.31*** | 0.37 | -0.06 | 64.4 | -0.28 | 0.64 | -0.91 |
| Early wet regrowth | 7 | 80 | 361.0 | 0.03*** | 0.99 | -0.96 | 345.8 | -0.32 | 1.80 | -2.12 |
| Wet season site prep | 8 | 91 | 701.7 | 0.62*** | 0.99 | -0.37 | 914.4 | -0.20** | 1.67 | -1.88 |
| Dry season final bed prep. and cultivation | 9 | 90 | 0 | 0.29*** | 0.32 | -0.02 | 10.8 | 0.06** | 0.91 | -0.85 |

[revised manuscript text omitted]

Figure 1

[Figure]

Figure 2

[Figure]

[Figure]

Figure 4

[Figure]

Plate 1